

# Inter-technique validation of tropospheric slant total delays

**Michal Kačmařík[1], Jan Douša[2], Galina Dick[3], Florian Zus[3], Hugues Brenot[4], Gregor Möller[5], Eric Pottiaux[6], Jan Kapłon[7], Paweł Hordyniec[7], Pavel Václavovic[2], Laurent Morel[8]**

[1] Institute of Geoinformatics, VŠB – Technical University of Ostrava, Ostrava, The Czech Republic
[2] Geodetic Observatory Pecný, Research Institute of Geodesy, Topography and Cartography, Zdiby, The Czech Republic
[3] Helmholtz Centre Potsdam - GFZ German Research Centre for Geosciences, Potsdam, Germany
[4] Royal Belgian Institute for Space Aeronomy, Brussels, Belgium
[5] Department of Geodesy and Geoinformation, Vienna University of Technology, Vienna, Austria
[6] Royal Observatory of Belgium, Brussels, Belgium
[7] Institute of Geodesy and Geoinformatics, Wroclaw University of Environmental and Life Sciences, Wroclaw, Poland
[8] GeF Laboratory, ESGT – CNAM, Le Mans, France

*Correspondence to*: Michal Kačmařík (michal.kacmarik@vsb.cz)

**Abstract.** An extensive validation of line-of-sight tropospheric Slant Total Delays (STD) from Global Navigation Satellite Systems (GNSS), ray-tracing in Numerical Weather Prediction Models (NWM) fields and microwave Water Vapour Radiometer (WVR) is presented. Ten GNSS reference stations and almost two months of data from 2013, including severe weather events, entered the comparison. Seven institutions delivered their STDs based on GNSS observations processed using five software and eleven strategies. STDs from NWM ray-tracing came from three institutions using three different NWM models. Results show generally a very good mutual agreement among all solutions from all the techniques. The mean bias (over all stations) between the GNSS solution selected as reference, which did not use post-fit residuals in STDs, and all other GNSS solutions without post-fit residuals is -0.6 mm for STDs scaled in the zenith direction, and the corresponding mean standard deviation is 3.7 mm. Standard deviations of comparisons between GNSS a NWM ray-tracing solutions are typically 10 mm +/- 2 mm (scaled in the zenith direction), depending on the NWM model and the particular station considered. When comparing GNSS versus WVR STDs, standard deviations reached 12 mm +/- 2 mm, as scaled in zenith direction. Moreover, the influence of adding raw GNSS post-fit residuals, as well as residuals screened out of systematic effects, to STDs was studied. It was found that adding raw post-fit residuals always led to lower quality of GNSS STDs while the situation was not that straightforward after the post-fit residuals cleaning.

## 1    Introduction

Tropospheric Slant Total Delay (STD) represents the total delay that undergoes the GNSS radio-signal due to the neutral atmosphere along his path from a satellite to a ground receiver antenna. This total delay can be separated into the hydrostatic part, caused by the atmospheric constituents, and the wet part caused specifically by water vapour. By quantifying the total delay, and by separating the hydrostatic and wet parts, it is possible to retrieve the amount of water vapour in the atmosphere along the path followed by the GNSS signal.





During the processing of GNSS observations only the total delay in the zenith direction (Zenith Total Delay, ZTD) above the GNSS antenna can be estimated for each epoch or for a time interval. ZTDs from GNSS reference stations are operationally assimilated into Numerical Weather Models (NWM) for almost a decade (Benitt and Jupp, 2012; Mahfouf et al., 2015). In Europe, this activity is coordinated mainly in the framework of the EUMETNET EIG GNSS Water Vapour Programme (E-GVAP, 2005-2017, Phase I-III, http://egvap.dmi.dk). Many recent studies demonstrated a positive impact of the ZTD or Integrated Water Vapour (IWV) assimilation on precipitation weather forecasts, especially of the short-time ones (Vedel and Huang, 2004; Guerova et al., 2006; Shoji et al., 2009; Guerova et al., 2016). On the other hand, continuous developments in NWM forecasting and nowcasting tools, as well as increasing needs for better predictions of severe weather events, stress the demand of high-quality humidity observations with high spatial and high temporal resolutions. While ZTDs provide information in zenith directions above GNSS stations, linear horizontal tropospheric gradients give information about the first-order spatial asymmetry around the station. Besides, Slant Tropospheric Delays (STDs) can provide additional details about the horizontal asymmetry in the troposphere, more specifically in the directions from a receiver to all observed GNSS satellites. With the increasing number of GNSS systems and satellites, the atmosphere scanning will be more complete, hence gaining even more interest. Bauer et al. (2011) showed a positive impact of STD assimilation into the Mesoscale Model 5 (MM5) and Kawabata et al. (2013) demonstrated a significant advantage of assimilating STDs into a high-resolution model in case of forecasting local heavy rainfall event against the scenario of assimilating ZTDs only. Also, Shoji et al. (2014) and Brenot et al. (2013) showed promising techniques for prediction of severe weather events using advanced GNSS tropospheric products such as horizontal gradients and STDs. The GNSS tomography technique aiming at the three-dimensional reconstruction of the water vapour field (Flores et al., 2001) uses STDs as input data as well. Obviously, the quality of the tomography depends on both the accuracy of the STDs (Bender et al., 2009) and the observation geometry (Bender et al., 2011).

Validation of GNSS slant delays with independent measurements is not a new research topic. GNSS slant delays were validated against WVR measurements in Braun et al. (2001), Braun et al. (2002) and Gradinarsky (2002). First attempts to derive slant delays from NWM fields and to compare them with GNSS STDs were carried out by De Haan et al. (2002) and Ha et al. (2002). Additional effort in evaluation of GNSS slant delays using WVR and NWM data was done at GFZ Potsdam during last years. Bender et al. (2008) showed an existing high correlation within the three sources (GPS, WVR, NWM) of slant wet delays and tried to quantify the effect of removing multipath from GPS post-fit residuals using a stacking method. Deng et al. (2011) validated tropospheric slant path delays derived from single- and dual-frequency GPS receivers with NWM and WVR data. Shang-Guan et al. (2015) compared GPS versus WVR slant IWV values (SIWV) using a 184-day long dataset. They also analysed the influence of the elevation angle setting and the meteorological parameters (used for the conversion to IWV) on the comparison results. More recently, a validation of multi-GNSS slant total delays retrieved in real-time from GPS, GLONASS, Galileo and BeiDou constellation was presented by Li et al. (2015a) using WVR and NWM as independent techniques for the assessment. Using multiple GNSS constellations brought a visible advantage, not only in terms of number of available slants, but also in their higher accuracy and robustness.





Nevertheless, most of the so-far presented studies were limited to only a single strategy for obtaining GNSS STDs, and usually restricted to a limited set of stations and/or a relatively short time period. The main purpose of this study is an extensive comparison of various solutions from GNSS processing, NWM ray-tracing and WVR measurements using one common dataset, and also comparing results from collocated stations. The GNSS solutions evaluated in this work used five

different software and eleven strategies, and exploited the GNSS4SWEC benchmark dataset (Douša et al., 2016). Then, the paper studies the impact of various approaches on STD estimates and aims at finding the most suitable strategy for estimating the GNSS-based STDs with the highest precision possible.

Section 2 briefly introduces the validation study dataset, and Section 3 describes the process of retrieving GNSS STDs including an overview of the different GNSS solutions. Section 4 provides a description of STDs generated from NWMs,

and Section 5 summarizes WVR principals and WVR-based STD solutions. Section 6 introduces the methodology used in the validation of STDs, and Sections 7 and 8 study the results achieved at single GNSS reference stations and at closely collocated stations, respectively.

## 2    Experiment description

The presented work has been carried out in the context of the EU COST Action ES1206 "Advanced Global Navigation

Satellite Systems tropospheric products for monitoring severe weather events and climate (GNSS4SWEC)" (http://www.cost.eu/COST_Actions/essem/ES1206, 2013-2017). Three mutually cooperating Working Groups (WG) have been established to cover the proposed topics: 1) WG1: Advanced GNSS processing techniques, 2) WG2: GNSS for severe weather monitoring, and 3) WG2: GNSS for climate monitoring. Validation of STDs belongs mainly under WG1, which is besides other topics oriented toward the development of new advanced tropospheric products. The idea of preparing a

common benchmark dataset, which could serve efficiently for most planned activities, was designed in the beginning of the project, and the data were collected, cleaned, documented, reference products generated and assessed (Douša et al., 2016). The selected geographical area is situated in central Europe (Austria, Germany, the Czech Republic, Poland) where severe weather events, including extensive floods on Danube, Moldau and Elbe rivers, occurred between May and June 2013. The benchmark dataset gathers observations from 430 GNSS reference stations, 610 meteorological synoptic stations, 21

radiosonde launching sites, 2 Water Vapour Radiometers (WVR), 2 meteorological radars, and output fields from the ALADIN-CZ Numerical Weather Prediction (NWP) model over a period of 56 days. ZTDs and horizontal tropospheric gradients from the reference GNSS and NWM-derived tropospheric products were already evaluated, and all resulted in very good agreements (Douša et al., 2016). All STDs used in this paper were computed by exploiting the benchmark dataset.

From the complete benchmark dataset, we selected a subset of 10 GNSS reference stations situated at six different locations

(Table 1). The selection was based on the following requirements: 1) long-term quality of observations and its stability, 2) availability of another GNSS reference station in the site vicinity, 3) availability of another instrument capable of STD measurements in the site vicinity, and 4) the location of the station w.r.t. its altitude and the weather events which occurred





during the evaluation period. The subset also includes collocated (dual) GNSS stations playing an important role in the validation. The collocated stations observed GNSS satellites with the same azimuth and elevation angles, so that they should theoretically deliver the same or very similar tropospheric parameters – ZTD, linear horizontal gradients and slant delays. Post-fit residuals of carrier-phase observations at the collocated stations should represent common effects due to the local

tropospheric anisotropy, while systematic differences could remain due to instrumentation and environmental effects such as antenna and receiver characteristics, and multipath. Only STDs from the WVR at Potsdam, collocated with the GNSS stations POTM and POTS, were available for this study because the second WVR, located at Lindenberg and collocated with the GNSS stations LDB0 and LDB2, was operating only in the zenith direction during the period of the study.

**Table 1**: Characteristics of 10 GNSS reference stations.

| Name | Latitude [°] | Longitude [°] | Height [m] | Network | Dual station | Receiver | Antenna |
|---|---|---|---|---|---|---|---|
| GOPE | 49.914 | 14.786 | 593 | IGS, EPN | | TPS NET-G3 | TPSCR.G3 TPSH |
| KIBG | 47.449 | 12.309 | 877 | | | TPS GB-1000 | TPSCR3_GGD CONE |
| LDB0 | 52.210 | 14.118 | 160 | | LDB2 | JAVAD TRE_G2T | JAV_GRANT-G3T NONE |
| LDB2 | 52.209 | 14.121 | 160 | | LDB0 | JPS LEGACY | LEIAR25.R4 LEIT |
| POTM | 52.379 | 13.066 | 145 | | POTS | JAVAD TRE_G3TH | JAV_GRANT-G3T NONE |
| POTS | 52.379 | 13.066 | 144 | IGS, EPN | POTM | JAVAD TRE_G3TH DELTA | JAV_RINGANT_G3T NONE |
| SAAL | 47.426 | 12.832 | 796 | | | TPS GB-1000 | TPSCR3_GGD CONE |
| WTZR | 49.144 | 12.879 | 666 | IGS, EPN | WTZS, WTZZ | LEICA GRX1200+GNSS | LEIAR25.R3 LEIT |
| WTZS | 49.145 | 12.895 | 663 | IGS | WTZR, WTZZ | SEPT POLARX2 | LEIAR25.R3 LEIT |
| WTZZ | 49.144 | 12.879 | 666 | IGS | WTZR, WTZS | JAVAD TRE_G3TH DELTA | LEIAR25.R3 LEIT |

# 3    Slant Total Delay retrievals from GNSS observations

The tropospheric Slant Total Delay (STD) cannot be estimated directly from GNSS data since the total number of unknown parameters in the solution would be higher than the number of observations. Instead, the total delays in the zenith direction above the GNSS station (i.e. ZTD) are adjusted, and optionally along with the total tropospheric linear horizontal gradients

($G$) to account for the first-order asymmetry of the local troposphere. The estimates are valid for individual processing epochs whenever using a stochastic approach, or for a given time interval when modelling the troposphere with a deterministic process, e.g. by piece-wise constant or linear models.

In practice, the ZTD is decomposed into an a priori model, usually by introducing the Zenith Hydrostatic Delay (ZHD, see Saastamoinen, 1972), and the estimated corrections, representing (mainly) the Zenith Wet Delay (ZWD). Similarly, the STD

is decomposed based on the *ZHD*, *ZWD* and *G* as described in Eq. (1) (Teke et al., 2011), where *ele* is the elevation angle and *azi* is the azimuth angle in degrees. The STD value is given in meters.



$$STD(ele, azi) = ZHD \cdot mf_h(ele) + ZWD \cdot mf_w(ele) + G(ele, azi) \qquad (1)$$

The elevation angle dependency of STD is described by the mapping functions, separately for the hydrostatic ($mf_h$) and the wet ($mf_w$) components. Nowadays, the Vienna Mapping Function (VMF1, see Böhm et al., 2006a) - or VMF1 like concepts -

is commonly used in GNSS data processing. Also, the empirical mapping function 'Global Mapping Function' (GMF, see Böhm et al., 2006b) is popular since it is consistent with VMF1 and easier to implement (independent on external data needing updates). Both, the VMF1 and the GMF are applicable down to 3° elevation angles.

The first-order horizontally asymmetric delay *G(ele,azi)* in Eq. (1) reflects local changes in temperature and particularly in water vapour. MacMillan (1995) proposed a model describing the gradient delay as a function of the elevation and azimuth

angles:

$$G(ele, azi) = mf_g \cdot (G_N \cdot cos(azi) + G_E \cdot sin(azi)) \qquad (2)$$

where $mf_g(ele) = mf_h(ele) \cdot cot(ele)$. Chen and Herring (1997) replaced the elevation dependent term $mf_h(ele) \cdot$

$cot(ele)$ by the gradient mapping function $mf_g(ele) = 1/(sin(ele) \cdot tan(ele) + C)$, with $C = 0.0032$, nowadays used as standard in GNSS data processing (Herring 1992). Typical range for $G_N$ and $G_E$ is from 0 mm to 2 mm however can reach up to 7 mm during a significant weather event. 1 mm corresponds to about 55 mm when projected to 7° elevation angle using the gradient mapping function.

For the analysis of GNSS L1 and L2 carrier-phase observations, a least-squares adjustment or a Kalman-filter approach was

applied to estimate the ZWDs and the two horizontal gradient components $G_N$ and $G_E$ at each GNSS site (Table 1) and for a specific validity period. Afterwards, Eq. (1) was used to compute STDs for each satellite in view. In addition, the observation post-fit residuals were stored since they might contain un-modelled tropospheric effects not covered by the estimated tropospheric parameters. These remaining effects are supposed to be mainly caused by high spatial and temporal variations of the humidity in the troposphere. Unfortunately, other un-modelled effects like multipath or satellite clock errors

can superimpose any tropospheric asymmetry information. Hence, the usage of the information content from the post-fit residuals for the reconstruction of the STDs remains an open question and it is further analysed in Sections 7 and 8.

Seven institutions delivered their STD solutions for this validation study, namely Ecole Supérieure des Géomètres et Topographes (ESGT CNAM), Geodetic Observatory Pecný (GOP, RIGTC), Helmholtz Centre Potsdam - German Research Centre for Geosciences (GFZ), Royal Observatory of Belgium (ROB), Technical University of Ostrava (TUO), Vienna

University of Technology (TUW), and Wroclaw University of Environmental and Life Sciences (WUELS). Principal information about individual solutions are given in Table 2 with a few specific notes important for the interpretation of the results.



GOP delivered two solutions based on the Precise Point Positioning (PPP) technique (Zumberge et al., 1997) and using the in-house developed application Tefnut (Douša and Václavovic, 2014) derived from the G-Nut core library (Václavovic et. al., 2013). Considering all available GNSS solutions, only GOP used a stochastic modelling approach to estimate the parameters. Additionally, GOP provided two solutions: 1) GOP_F using Kalman filter (forward filter only), i.e. capable of

providing ZTD, tropospheric gradients and STDs in real time, and 2) GOP_S applying the backward smoothing algorithm (Václavovic and Douša, 2015) on top of the Kalman filter in order to improve the quality of all estimated parameters during the batch processing interval and for avoiding effects such as the PPP convergence or re-convergence.

Some institutions delivered also two STD solutions in such a way that these solutions only differ in a single processing setting. The aim was to evaluate their impact on STDs: a) TUO_G and TUO_R exploit GPS-only and GPS+GLONASS

observations respectively, b) TUW_3 and TUW_7 apply an elevation cut-off angle of 3 and 7 degrees respectively, and c) ROB_G and ROB_V use the GMF and VMF1 mapping functions respectively. Additionally, ROB solutions are the only ones based on the processing of double-difference (DD) observations and providing zero-differenced (ZD) carrier-phase post-fit residuals converted from the original DD residuals using the technique described in Alber et al. (2000).

In total, we compared eleven solutions computed with five different GNSS processing software. Five of the solutions used

GPS and GLONASS observations and six solutions used GPS-only observations; five of them are based on double-difference observations and six of them are computed using zero-difference data in PPP analysis. More information about TUW solutions can be found in Möller et al. (2016), about GFZ in Bender et al. (2009, 2011), Deng et al. (2011) and about CNAM in Morel et al. (2014). For ROB, TUO and WUE solutions we refer the reader to Dach et al. (2015).

Whenever ZD post-fit residuals were available for any solution, three variants of the solution are presented in the paper: 1)

solution without residuals (nonRES), 2) solution with raw residuals (rawRES), and 3) solution with cleaned residuals (clnRES). The cleaning of the post-fit residuals used to eliminate systematic effects such as multipath and antenna phase centre variations was done via generating time-/azimuth-/zenith-dependent residuals correction maps as described by Shoji et al. (2004). For each solution and each station, the mean of the post-fit residuals in 1×1 degree bins were computed over the whole benchmark period while residuals exceeding ±3 times the standard deviation were excluded from the computation of

mean. Computed means were then subtracted from the original post-fit residuals to generate the variant introducing cleaned residuals.



**Table 2**: Information about individual GNSS-based STD solutions used in the validation.

| Solution Name | Institution | Strategy | Software | GNSS | Elev. cut-off | Mapping function | Products | ZTD/gradients interval | ZD post-fit residuals |
|---|---|---|---|---|---|---|---|---|---|
| CNAM | ESGT CNAM | DD | GAMIT | GPS | 3 ° | VMF1 | IGS final | 1h / 1h | NO |
| GFZ | GFZ Potsdam | PPP | EPOS 8 | GPS | 7 ° | GMF | GFZ | 15min / 1h | YES |
| GOP_F | GO Pecný | PPP | G-Nut/Tefnut | GPS | 7 ° | GMF | IGS final | 2.5min / 2.5min | YES |
| GOP_S | GO Pecný | PPP | G-Nut/Tefnut | GPS | 7 ° | GMF | IGS final | 2.5min / 2.5min | YES |
| ROB_G | ROB | DD | Bernese 5.2 | GPS+GLO | 3 ° | GMF | CODE final | 15min / 1h | YES |
| ROB_V | ROB | DD | Bernese 5.2 | GPS+GLO | 3 ° | VMF1 | CODE final | 15min / 1h | YES |
| TUO_R | TU Ostrava | DD | Bernese 5.2 | GPS+GLO | 3 ° | VMF1 | CODE final | 1h / 3h | NO |
| TUO_G | TU Ostrava | DD | Bernese 5.2 | GPS | 3 ° | VMF1 | CODE final | 1h / 3h | NO |
| TUW_3 | TU Vienna | PPP | NAPEOS | GPS+GLO | 3 ° | GMF | ESA final | 30min / 1h | YES |
| TUW_7 | TU Vienna | PPP | NAPEOS | GPS+GLO | 7 ° | GMF | ESA final | 30min / 1h | YES |
| WUE | WUELS | PPP | Bernese 5.2 | GPS | 3 ° | VMF1 | CODE final | 2.5min / 1h | YES |

## 4    Computation of Slant Total Delay from Numerical Weather Prediction model

Simulating STDs in NWP models consists in integrating the atmospheric refractivity through the path followed by GNSS signals. STDs have been simulated using three different NWMs: ALADIN-CZ (4.7 km-resolution limited-area hydrostatic model), ERA-Interim (1° horizontal resolution), and NCEP-GFS (1° horizontal resolution). For more details, see Douša et al. (2016). First, STD solutions using the ERA-Interim and NCEP-GFS models were delivered by GFZ Potsdam using acronym ERA/GFZ and GFS/GFZ, respectively. Only a short introduction is provided in Section 4.1 since the GFZ tool for an accurate and ultra-fast NWM ray tracing has been described in other papers cited below. Two STD solutions were then delivered for the ALADIN-CZ model: a) the ALA/BIRA, which was generated at Royal Belgian Institute for Space Aeronomy (BIRA), described in Section 4.2, and b) the ALA/WUELS, which was delivered by Wroclaw University of Environmental and Life Sciences, described in Section 4.3.

### 4.1    Description of ERA-Interim STD solution (ERA/GFZ) and NCEP-GPS STD solution (GFS/GFZ)

The ERA-Interim and NCEP-GFS STD solutions by GFZ are based on 'assembled' STDs. At first, for the considered station and epoch, a set of ray-traced STDs (various elevation and azimuth angles) is computed using technique described in Zus et al. (2014). Secondly, from this set of ray-traced STDs, the tropospheric parameters (i.e. zenith delays, mapping function coefficients, first- and higher-order gradient components) are determined. Finally, for the required azimuth and elevation angle the STD is 'assembled' using the tropospheric parameters. For a detailed description of the tropospheric parameter determination the reader is referred to Douša et al. (2016). The differences between the 'assembled' and ray-traced STDs are sufficiently small in particular for elevation angles above 10° (Zus et al., 2016). In essence, the largest uncertainty in the 'assembled' (and ray-traced) STDs remains the uncertainty of the underlying NWM refractivity field. This uncertainty is estimated to be about 8-10 mm close to the zenith increasing to about 8-10 cm at an elevation angle of 5 ° (Zus et al., 2012). Similar uncertainty of around 8 mm for the zenith direction was also found for ALADIN-CZ model  in Douša et al. (2016).





## 4.2 Description of ALADIN-CZ STD solution from BIRA (ALA/BIRA)

To compute STDs from ALADIN-CZ, a simplified strategy has been used to model the curve path followed by GNSS signals through the neutral atmosphere, as suggested by Saastamoinen (1972). The delays simulated with this strategy show small differences in comparison to straight line simulations (differences of about 4 mm, 5 mm and 10 mm respectively at
15°-, 10°- and 5°-elevation). Simulations have computed STDs down to 3°-elevation, however under an elevation of 15°, a proper ray tracing strategy, as mentioned in Section 4.1 should be preferably applied.

For each latitude-longitude grid point and each level of ALADIN-CZ model, the NWP outputs considered to compute STDs are: geopotential height (*geopotH* in m), pressure (*P* in Pa), temperature (*T* in K), partial pressure of water vapour (*e* in Pa), mix ratio of liquid and solid water (in kg/kg). The ground pressure of each column is also retrieved. The geopotential height
is converted to the altitude above the geoid:

$$h_{geoid} = (g_0 * R_e * geopotH) / (g * R_e - g_0 * geopotH) \qquad (3)$$

where $g_0$ is a standard gravity acceleration (mean value of *9.80665 m/s²* from the World Meteorological Organization, WMO); $R_e = 6378137 / (1.006803 - 0.006706 * sin² (lat))$ is the radius of the ellipsoid in meter for the latitude (*lat* in
degrees); *g* is the gravity acceleration (in m/s²) of the considered location given as:

$$g = 9.7803267714 * (1. + 0.00193185138639 * sin²(lat)) / \sqrt{1. - 0.00669437999013 * sin^2(lat)} \qquad (4)$$

Then, the height above the geoid is converted to height above the WGS84 ellipsoid (in m) with the use of the EGM96 (Earth
Gravitational Model, Lemoine et al., 1998) undulation. Note that for the region of the benchmark campaign the difference between geoid and WGS84 altitude is about 47 m.

Using the hypsometric equation, the ground pressure and the pressure of each level are considered to estimate the altitude for the different levels. In total ALADIN-CZ outputs provide 87 levels up to an altitude of about 55 km. However, to assess STDs from ALADIN-CZ, the integration was stopped at 15 km since the contribution of water vapour above this altitude is
negligible. An adaptive step is considered (100 m, 200 m, 250 m, 500 m, 1000 m, respectively for vertical altitudes between 0-1 km, 1-3 km, 3-5 km, 5-10 km, 10-15 km). Bi-linear interpolations of ALADIN-CZ parameters at the altitude of the GNSS station and for each step of the integration were proceeded. Note that there is no station selected for the validation located below the first layer of ALADIN-CZ.

The expression of simulated STDs from ALADIN-CZ is the summation of these four contributions:

$$STD = SHD_{int} + SWD_{int} + SHMD_{int} + STD_{ext} \qquad (5)$$





where $SHD_{int}$, $SWD_{int}$ and $SHMD_{int}$ are respectively the inside-model integration contribution of the hydrostatic, wet and hydrometeor delays, and $STD_{ext}$ is the external model contribution (over 15 km).

$$SHD_{int} = 10^{-6} \sum_{k=1}^{k=k_{top}} k_1 \frac{P_i}{Tv_i} \Delta s_i \qquad (6)$$

$$SWD_{int} = 10^{-6} \sum_{k=1}^{k=k_{top}} \left( k'_2 \frac{e_i}{T_i} + k_3 \frac{e_i}{T_i^2} \right) \Delta s_i \qquad (7)$$

with $k'_2 = k_2 - k_1 * R_d / R_w$ where $k_1$ (in K/Pa), $k_2$ (in K/Pa), $k_3$ (in K²/Pa) are the empirical refractivity coefficients of Bevis et al. (1994), $R_d$ and $R_w$ the gas constants respectively for dry air and water vapour (in J/kmol K), and $T_v$ is the virtual temperature (in K). For the estimation of the hydrometeor contribution inside the model, as presented in Eq. (8), ($N_{lw}$, $M_{lw}$) and ($N_{ice}$, $M_{ice}$) are (atmospheric refractivity, mass content per unit of air volume) of the liquid and ice water, respectively.

$$SHMD_{int} = 10^{-6} \sum_{k=1}^{k=k_{top}} (N_{lw} + N_{ice}) \Delta s_i = \sum_{k=1}^{k=k_{top}} (\alpha_{lw} M_{lw} + \alpha_{ice} M_{ice}) \Delta s_i \qquad (8)$$

The estimation of coefficients $\alpha_{lw} \sim 1.45$ and $\alpha_{ice} \sim 0.69$ is presented in Brenot et al. (2006). The ALADIN-CZ model provides mixing ratios of cloud water (liquid components) and pristine ice (solid water components). The mass content per unit of air volume is obtained using the associated mixing ratio, pressure, water vapour partial pressure and temperature. $STD_{ext}$ is obtained with the hydrostatic formulation (Saastamoinen, 1972) mapped with $mf_h$ (see Eq. 1) and using the elevation, latitude and pressure of the last step of the integration (i.e. at 15 km). Note that the wet contribution over 15 km is neglected since it is practically zero. The estimation of $STD_{ext}$ (about 0.21 m) provides sufficiently accurate modelling for the hydrostatic contribution over 15 km (as shown by the sensitivity test from Brenot et al., 2006).

### 4.3    Description of ALADIN-CZ solution from WUELS (ALA/WUELS)

The ray-traced tropospheric delays for WUELS solution are based on piece-wise bent-2d model propagation. Thus, it prevents to know the exact trajectory in advance in contrary to straight-line model and needs to be solved iteratively based on preceding ray refractive index. Similar examples are given by Böhm and Schuh (2003) or Hobiger et al. (2008). We assume the ray-path does not leave the plane of constant azimuth for a given elevation angle to a satellite. The out-of-plane contribution to the delay is thus neglected making the propagation two-dimensional, hence 2d. The real ray-path is then approximated by a finite number of linear ray-pieces in WGS84 coordinates using Euler formula for Earth radius

$$R = (cos^2A / M + sin^2A / N)^{-1} \qquad (9)$$



where $A$ is the azimuth angle between a satellite and a receiver, $M$ and $N$ are radii of curvature along meridian and prime vertical, respectively. We follow height-dependent increments as presented in Rocken et al. (2001): 10 m, 20 m, 50 m, 100 m, 500 m respectively for geometric altitudes between 0-2 km, 2-6 km, 6-16 km, 16-36 km, and above 36 km which require meteorological parameters to be vertically interpolated in order to obtain finer resolution. Both $P$ and $e$ are interpolated

exponentially from two nearest layers, while the temperature change is considered linear. Horizontally, we find the four nearest nodes for each ray to perform weighted mean interpolation, where the weighting function equals to the inverse squared distance. The reference hybrid-level of the ALADIN-CZ model is determined by surface geopotential which is converted to geopotential meters by dividing the geopotential values by $g_0$. Meteorological parameters are expressed on pressure levels which represent standard vertical coordinate. The hypsometric equation is used to calculate geometric

thickness between consecutive isobaric surfaces

$$dz = R_d * T_m / g_0 * ln(P_1 / P_2) \qquad (10)$$

where $R_d = 287.058 \, J/K/kg$ is the gas constant for dry air, $T_m$ is the mean virtual temperature of the layer between $P_1$ and $P_2$

pressure levels in Kelvin. The conversion from ALADIN-CZ vertical coordinate system to geometric altitudes is then consistent with the BIRA approach described in Section 4.2. In WUELS solution, the signal tracking is performed exploiting a full model vertical resolution up to the uppermost ALADIN-CZ layer at 55 km. Above the top layer, we adopt the U.S. Standard Atmosphere (1976) to provide supplementary meteorological data up to 86 km. For each ray-path coordinates, the refractive index is calculated as a function of $P$ (in hPa), $e$ (in hPa) and $T$ (in K) with empirically derived "best available"

coefficients $k$ given by Rueger (2002).

$$N = (n - 1) \times 10^6 = k_1 * (P - e)/T + k_2 * e/T + k_3 * e/T^2 \qquad (11)$$

The contribution of water droplets and ice crystals in the atmosphere is neglected in the total delay. All tropospheric delays

are traced with respect to vacuum elevation angles. The electromagnetic delay is calculated for a given chord length ($s$) using the mean refractive index $n$ between two consecutive rays yielding the total delay in meters

$$STD = \sum s_i(n_i - 1) \times 10^6 \qquad (12)$$

which can be separated on hydrostatic and wet part using respective refractive indices. Additionally, to the radio path length, the accumulated bending effect (bend) along the ray path is added to the hydrostatic mapping function which, together with the wet mapping function, can be calculated as follows:

$$bend = \sum (s_i - cos(ele_i - ele_k) \, s_i) \qquad (13)$$





$$mf_h = (SHD + bend) / ZHD \qquad\qquad (14)$$

$$mf_w = SWD / ZWD \qquad\qquad (15)$$

where $ele_i$ is the elevation angle for a given model layer and $ele_k$ is the outgoing elevation angle at uppermost altitude.

### 4.4    Assessment of the Hydrostatic, Wet and Hydrometeor Contributions to the Slant delays

5    During the whole period of the benchmark campaign, the maximum contribution of hydrometeors reached 17 mm during the extreme weather events on 20-23 June (Douša et al., 2016). The 2D fields of ZTD, ZHD, ZWD, and ZHMD (zenith hydrometeor delays) are presented in Figure 1. They illustrate the large-scale convection with the presence of hydrometeors along the convergence line associated with a strong contrast of dry and wet air masses. The contribution of hydrometeors to ZTD reached up to 7 mm (as scaled in the zenith direction) for the stations POTS and POTM at 15:00 UTC on 23 June 2013

10    (see Figure 1d). According to satellite trajectories at this time for the station POTS, a maximum SHMD of 25.6 mm is observed for a satellite at 22°-elevation angle.

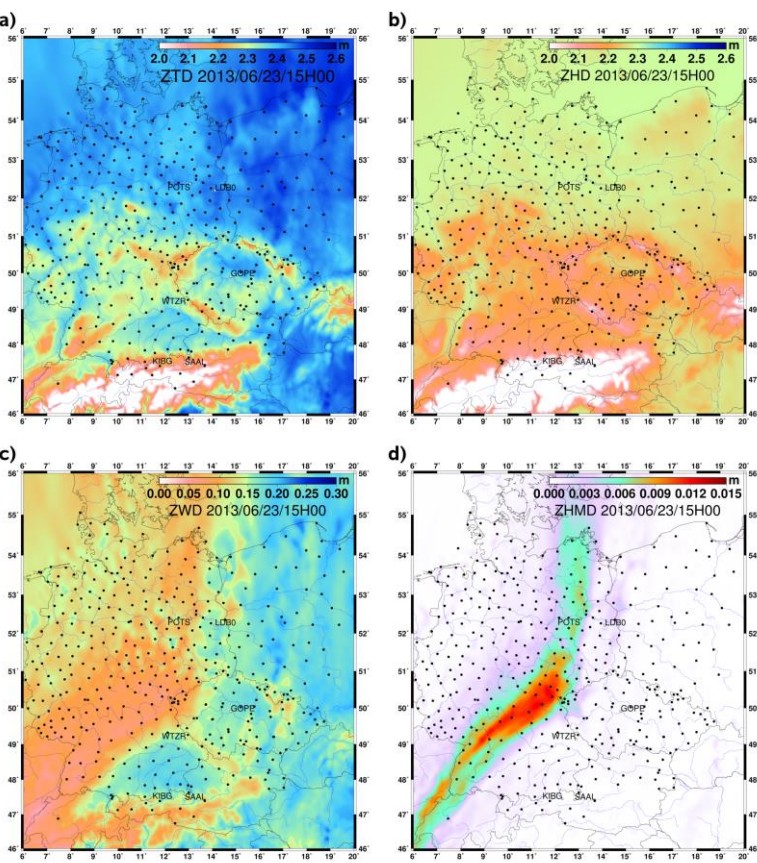

Figure 1: Simulation of ZTD, ZHD, ZWD and ZHMD at 15:00 UTC on 23 June 2013. Each black dot represents a GNSS station included in the benchmark dataset. For stations included in this STD validation study their names are given.



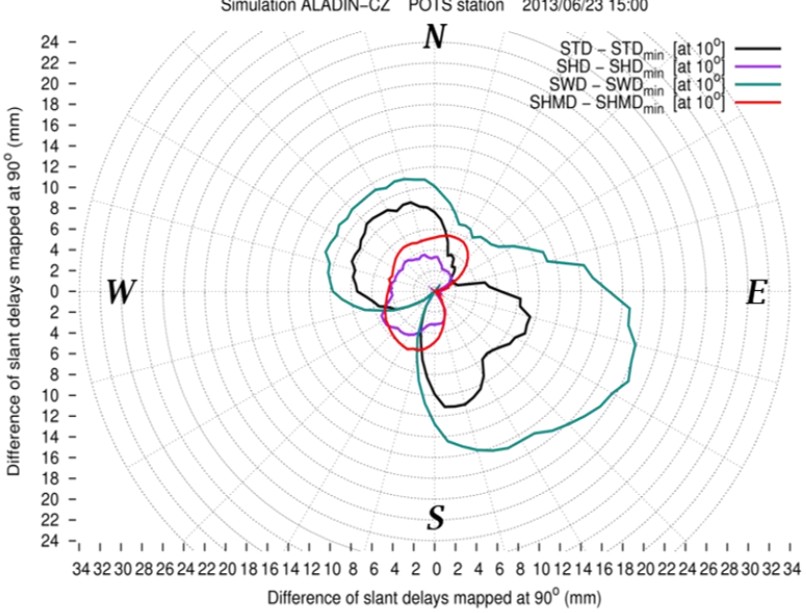

Figure 2: Skyplot of differential slant delays simulated at 10° and mapped at 90°, for a 360° azimuthal range (at 15:00 UTC on 23 June 2013). For total, hydrostatic, wet and hydrometeors delays, a differential slant delay is the difference between a slant delay simulated and the respective minimum value (obtained considering slant delays simulated at 10° elevation along all the azimuthal directions).

Figure 2 shows simulated STDs for a cone with a 10° elevation angle during the severe weather condition of the 23$^{rd}$ of June 2013, and mapped in the zenith direction (at 90°) using the mapping functions of Eq. (1): $mf_h$ for SHD and $mf_w$ for SWD and SHMD. For this 10°-cone, the minimum present values of total, hydrostatic, wet and hydrometeors delays simulated at 15:00 UTC on 23 June, are given as STDmin, SHDmin, SWDmin and SHMDmin in Figure 2. The respective differences of STD, SHD, SWD and SHMD and corresponding minimum values simulated at 15:00 UTC are presented in Figure 2. The anisotropic variation of total, hydrostatic, wet and hydrometeor delays can be visualised on a skyplot. As a confirmation of Figure 1b and 1d, Figure 2 shows weak hydrostatic anisotropy. This anisotropy (up to 5.8 mm) is almost the same as the hydrometeors one (up to 6 mm). The area within the red curve is larger than the purple area (hydrostatic anisotropy), meaning that the total effect of the hydrometeor anisotropy is slightly larger than the one from the hydrostatic component. Note that Figure 2 shows the anisotropies simulated at 10° and mapped at 90° (giving an idea of the variations in the zenith direction). The largest anisotropy is clearly induced by water vapour (values up to 20 mm in the south-east direction of POTS, also shown in Figure 1c). With mean hydrostatic and hydrometeor anisotropies oriented in the opposite direction of the wet one, Figure 2 presents a total anisotropy with weaker values (up to 12 mm) than the wet anisotropy.



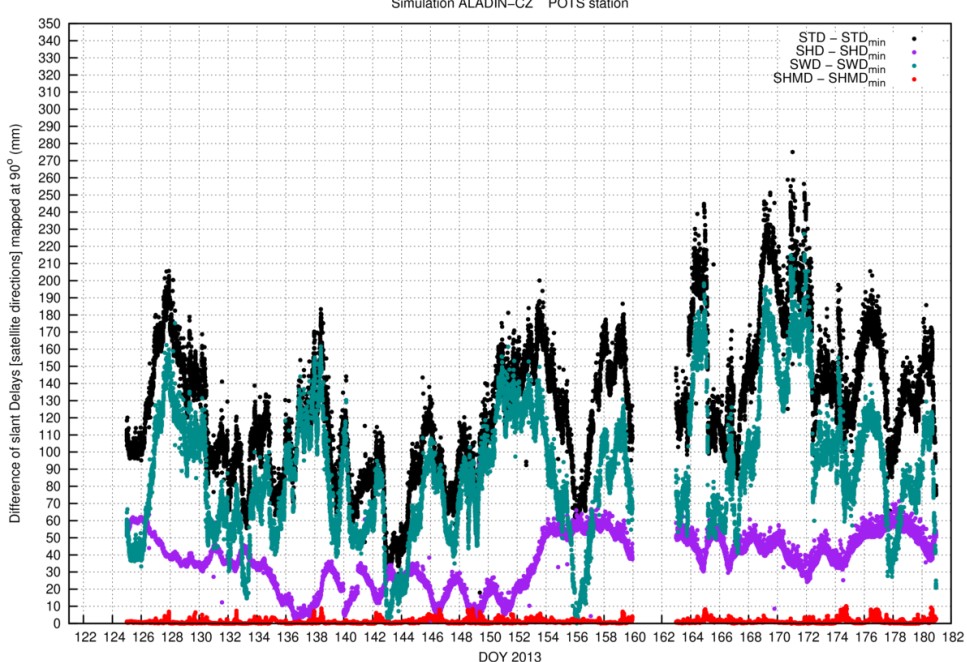

Figure 3: Time-series of slant delays (STD, SHD, SWD, and SHMD) differences (in direction of all GNSS visible satellites, then mapped in the zenith direction) during the whole period of the benchmark campaign for the station POTS.

To complement the snapshot of Figure 2 the time-evolutions of SHD, SWD, SHMD, and STD in the direction of all observed GNSS satellites for the station POTS are presented in Figure 3. Slant delays have been simulated in the direction of observed satellites (hydrostatic, wet, hydrometeor and total contributions) and corresponding delays in the zenith direction have been computed and mapped using mapping functions presented in Eq. 1 ($mf_h$ for SHD and $mf_w$ for SWD and SHMD). These values ($STD_{min}$ = 2310.6 mm, $SHD_{min}$ = 2240.8 mm, $SWD_{min}$ = 43.1 mm and $SHMD_{min}$ = 0 mm) have been subtracted from their corresponding values simulated in direction of satellites. Then, the differences have been mapped back at 90°. For day of year (DOY) 174 (i.e. 23$^{rd}$ of June 2013), we can see a contribution of hydrometeors up to 10 mm. Looking at the whole period of the benchmark campaign, the variation range of STD, SHD and SWD (mapped at 90°) are 275 mm, 80 mm and 230 mm, respectively. Figure 3 confirms the interest of GNSS delay observations for meteorology and the detection of variation of water vapour, pointing the importance of simulating slant total delays (that take into account mainly the variation from wet delay, but also from hydrostatic delays and occasionally from hydrometeors in specific severe weather cases). Note that there is no data available for POTS station between DOY 121 and 125 and DOY 160 and 163. For this reason, we have not simulated the slant delays for this period, as shown by the gaps in Figure 3. The simplified strategy used to simulate curve slant paths gives some inaccurate simulations of slant delays for elevations between 3° and 5°, shown by isolated values in Figure 3. Such inaccuracies could be avoided by using a ray-tracing algorithm. For a comprehensive overview on ray-tracing algorithms and comparisons the reader is referred to Nafisi et al. (2012).





## 5    Water Vapour Radiometer Measurements

During the benchmark period, the WVR located at GFZ Potsdam operated in a mode scanning the atmosphere at selected elevation and azimuth angles. The instrument is situated on the same roof as the GNSS reference stations POTM and POTS. All three devices are within ten meters from each other. The HATPRO WVR from Radiometer Physics was set up to scan the atmosphere to extract profiles of atmospheric temperature, water vapour and liquid water using frequencies between 22.24 and 27.84 GHz and a window channel at 31.4 GHz. The WVR is switching between 'zenith mode' when it is measuring Integrated Water Vapour (IWV) and 'slant mode' when it is tracking GPS satellites using an in-built GPS receiver. In the latter case, Slant Integrated Water Vapour (SIWV) values are delivered for the direction of satellites. Since the instrument can track only one satellite at one moment the number of observations is quite limited compared to slants from GNSS which are simultaneously observed from several GNSS satellites.

Our study focuses on the comparison of STDs, not SIWV. It was thus necessary to convert the WVR SIWV into STDs. Firstly, WVR observations with rain flag and Atmospheric Liquid Water (ALW) values exceeding 1 kg*m$^{-2}$ were rejected. Both rain and high values of ALW can significantly distort the quality of WVR measurements. Secondly, SIWV values were converted into Slant Wet Delays (SWD) using the Askne and Nordius (1987) formula and the refractivity constants from Bevis et al. (1994). ZHD values were computed with the precise model given by Saastamoinen (1972). For the described conversions, we used values of the atmospheric pressure and temperature measured in situ of the GNSS reference station POTS. A hydrostatic correction for the altitude difference between the meteorological station and the WVR position was applied to the atmospheric pressure values. ZHD values were mapped to elevation angles of the WVR using the hydrostatic mapping function derived from the NCEP-GFS. In order to convert accurately SIWV to STDs, we took into account the influence of the hydrostatic horizontal gradients (see e.g. Li et al., 2015b). We used the hydrostatic horizontal gradients derived from the NCEP-GFS for that purpose. Finally, SHD and SWD values were summed up to deliver STDs. The described conversion of WVR SIWVs to STDs aimed at minimum distorting the accuracy of original WVR observations.

## 6    Methodology of STD comparisons

We provide the specificities of each type of technique comparisons in this section. Since NWM outputs are restricted to the time resolution of their predictions (typically one, three or six hours) and since WVR is able to track only one satellite at one moment, all three sources provide different numbers of STDs per day. Therefore, three different comparisons are presented: 1) results for GNSS versus GNSS comparisons, 2) results for GNSS versus NWM comparisons, and 3) results for GNSS versus WVR comparisons. Section 7 presents the validation at individual stations and Section 8 inter-compare results obtained at GNSS dual stations. All the given results are obtained over the whole benchmark period. No outlier detection and removal procedure was applied during the statistics computation within the study.

Two variants of the comparisons are always presented: 'ZENITH' and 'SLANT'. 'ZENITH' stands for original STDs mapped back to zenith direction using *1/sin(e)* formula. This mapping aimed at normalizing STDs for the evaluation as a





whole unit. The 'SLANT' type of comparison denotes an evaluation of STDs at their actual elevation angles. To be more specific, slant delays were grouped into individual elevation bins of 5 degrees, i.e. for example all slants with an elevation angle between 10 and 15 degrees were evaluated as a single unit. There was one exception regarding the size of a bin since the lowest one contained slants from 7° to 10° elevation angle, 7° being the lowest elevation angle common to all GNSS

STD solutions. This cut-off angle was thus used in all GNSS versus GNSS and GNSS versus NWM comparisons.

Presented values of bias and standard deviations were always computed directly from all STDs within the processed benchmark campaign period, therefore they are not based on any kind of daily or other averaging. In some tables, only median values of bias and standard deviation over all GNSS STD solutions (Tables 7, 9 and 10) or over all processed stations (Tables 3 and 4) are given to consolidate the presentation of validation results.

## 6.1    GNSS versus GNSS comparisons

In the case of individual inter-GNSS solutions validation, the situation was straightforward and no interpolation nor specific hypothesis was necessary: the comparisons were done on a direct point-to-point basis of observations coming from identical azimuth and elevation directions.

## 6.2    GNSS versus WVR comparisons

To find pairs of STDs observations between WVR and GNSS, the following rules were used: 1) the time difference between both observations had to be shorter than 120 s, and 2) the difference between both azimuth and elevation angles had to be smaller than 2.5° and 0.25°, respectively. From these criteria, the maximum difference in elevation angle has the largest impact on the number of observation pairs found. Hence the smaller values for these settings, the smaller number of pairs found and the higher standard deviations resulted between GNSS and WVR STDs. As an illustration, a change from 0.35° to

0.25° led to the decrease of the number of STD pairs between the GNSS GFZ solution and the WVR at station POTS from 63,703 to 48,583 pairs; the standard deviation of the projected STD differences in the zenith direction then decreased from 14.6 to 11.7 mm too. Since the bias practically remained unchanged (-6.1 mm versus -5.9 mm), the applied selection procedure mainly influenced the stability of the comparison between WVR and other sources of slant delays. When comparing GNSS versus WVR STDs, a cut-off elevation angle was set to 15 degrees to exclude low-elevation angle

observations from WVR as their quality could be further degraded by a ground radiation or other local environment conditions.

## 6.3    GNSS versus NWM comparisons

Given the very small distances between collocated antennas and the coarse resolution of the global NWM models, STDs from NWM ray-tracing using the ERA-Interim and the NCEP GFS models were derived only for one of the collocated

stations. The same set of NWM-derived STDs was then used for the validation of the results at the collocated receivers.




## 7 Results at individual stations

### 7.1 GNSS versus GNSS

The total STD pairs available for this part of the validation is roughly 1.7 million, and varies from 140,987 to 206,320 according to the station. Individual GNSS solutions were first compared to the GFZ solution in the zenith direction

(ZENITH). We chose the 'GFZ' solution as reference because GFZ Potsdam has long-term experience in producing GPS slant delays and because the GFZ near real-time solution for German GNSS reference stations is already being operationally delivered to the Deutscher Wetterdienst - The German Meteorological Service for NWM assimilation testing purposes (Bender et al., 2016). Figure 4 shows all the solutions using STDs calculated from the estimated ZTD and horizontal gradient parameters, i.e. without adding post-fit residuals. Adding raw or clean residuals, consistently to both compared and reference

solutions, provided very similar graphs (not displayed). Colours in Figure 4 indicate the processing software used in individual solutions. Medians of all solutions (dotted lines in each bin) are displayed for each station in order to highlight differences among the stations. These were observed mainly as systematic errors ranging from -3.6 mm to 0.6 mm. The better agreement between GOP and GFZ solutions could be attributed to a similar strategy of both solutions compared to others. It is particularly visible for LDB0 and POTM stations where median values over all solutions differ by -2.3 mm and -

3.6 mm, respectively. The reason for the divergent behaviour at the two stations has not been identified although site metadata were cross-checked carefully. A significant difference can also be noticed for TUW_3 and TUW_7 at the station KIBG where these solutions used individual antenna calibration files while all others solution used type mean calibration (Schmid et al., 2016). On the other hand, plots with standard deviations show agreements within 3-5 mm among all the stations and all solutions. The only exception is the GOP_F solution representing a simulated real-time analysis applying

only a Kalman filter (not backward smoothing) and providing results by a factor of 2 worse compared to the others in terms of precision.

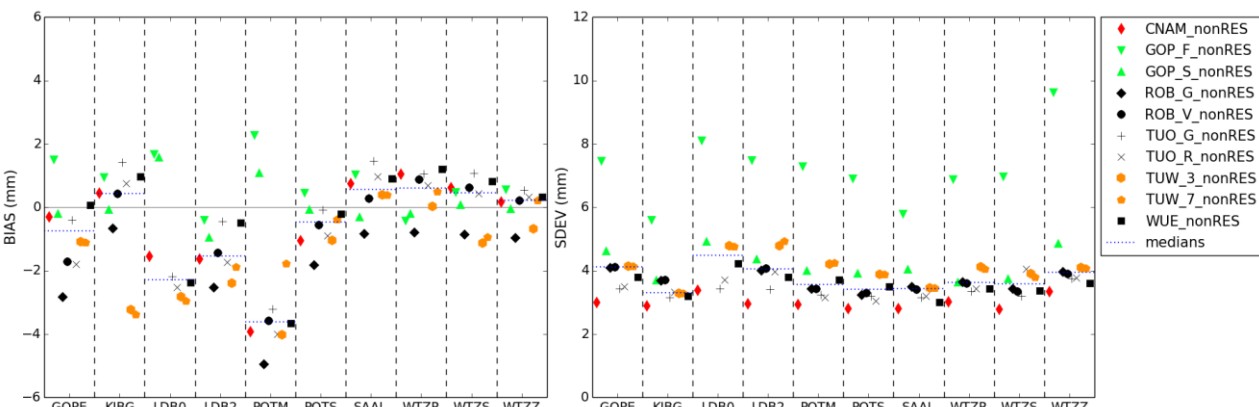

Figure 4: Comparison of individual GNSS STD solutions against GFZ solution, all without using residuals (nonRES) and projected in the zenith direction: bias (left) and standard deviation (right). The median value of all solutions at each station is

represented by the dotted blue line in each bin.



All individual GNSS STD solutions were then compared independently using none (nonRES), raw (rawRES) and clean (clnRES) residuals. The comparison aimed at assessing the impact of different strategies for reconstructing GNSS STDs. Figure 5 displays biases and standard deviations for all solutions when comparing STDs with and without raw residuals. Similarly, Figure 6 shows results for STDs with and without clean residuals. Both comparisons demonstrate systematic

errors at a sub-millimetre level over all stations and solutions. Smaller biases are however observed in the latter case (clnRES) which indicates the impact of station-specific systematic errors in raw residuals when projected into zenith directions. Although the decrease of biases is visible for all solutions, several solutions (GFZ, GOP, WUE) resulted with almost zero biases over all the stations. It could be attributed to a better possibility of removing systematic effects in PPP as absolute residuals are accessible directly. This is in contrast to the solutions using double-difference observations (ROB)

which need to reconstruct residuals from double-difference ones representing relative information only. Interestingly, the TUW PPP solutions seem to perform similarly to the ROB DD solution in this case.

Comparing standard deviations in both figures demonstrates that the impact of adding clean residuals on the precision reached from 2.5 mm to 4.5 mm, while adding raw residuals resulted in increased discrepancies between 3.0 mm and 6.5 mm. The station-specific behaviour is more obvious for the latter rather than for the former, and generally the relative

performance over all stations is in a good agreement among different solutions. In particular, LDB0 and LDB2 stations show a worse quality in performance which was observed already in Figure 4. Their standard deviations were however also significantly reduced after cleaning the residuals and became more homogeneous with other stations. In this context it should be noted that the station LDB0 is missing in both ROB solutions since it has been excluded from the network solution during the pre-processing phase due to a lower quality of observations. Besides the GOP_F specific solution, we can observe that

the GOP_S solution performs worse by about 25% compared to the other solutions. This can be attributed to the stochastic model using the epoch-wise parameter estimation and to potential deficiencies in implementations of all necessary GNSS models. The latter is considered because the software was the only one developed from scratch recently and, in contrast to all others, could not have been extensively used in a variety of applications. Finally, there are rather small differences observed due to the applied strategy, namely forward versus backward filtering, GPS versus GPS+GLO and the cut-off 3 versus 7

degrees for elevation angles (statistically compared for STDs above the elevation angle cut-off 7 degrees).





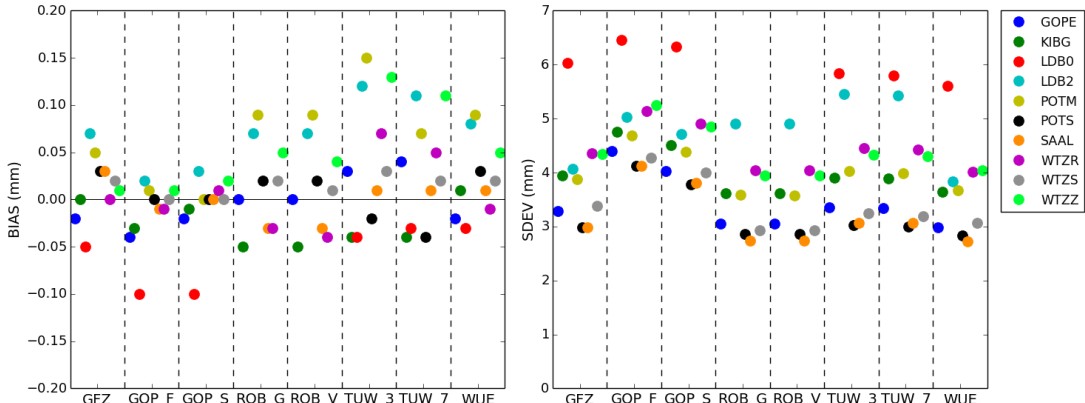

Figure 5: Comparison of individual GNSS STD solutions without residuals (nonRES) and with raw residuals (rawRES); statistics are projected in the zenith direction: bias (left) and standard deviation (right).

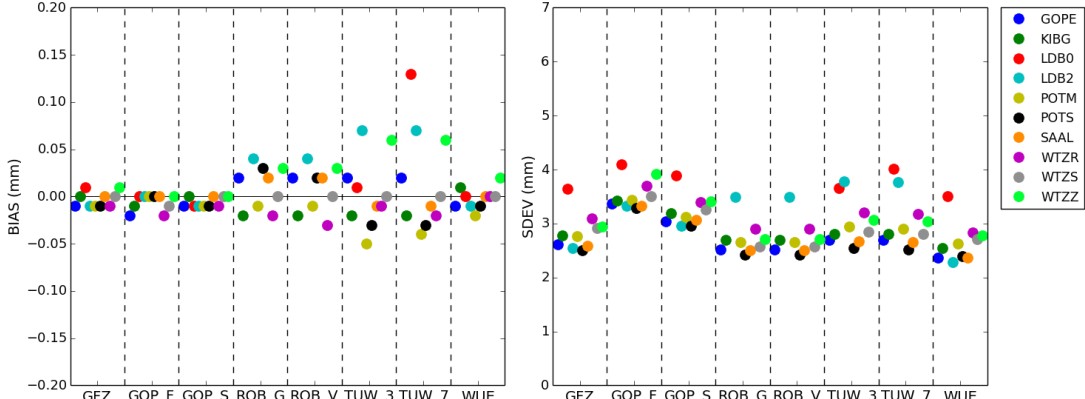

Figure 6: Comparison of individual GNSS STD solutions without residuals (nonRES) and with clean residuals (clnRES); statistics are projected in the zenith direction: bias (left) and standard deviation (right).

Table 3 summarizes statistics related to the figures providing medians and standard deviations over all stations. Notably, systematic errors of STDs (over all stations) expressed in the zenith direction are negligible in all solutions, i.e. not affected by adding raw or clean residuals. The impact of adding raw residuals to the estimated model can be characterized by the

median RMS of 3.9 mm (see the first two data columns in the table) which may vary strongly for different stations, as evident for stations LDB0 and LDB2 in Figure 5 and Figure 6. In this comparison, the use of clean residuals reached the impact of 2.8 mm (see the middle two data columns in the table) corresponding to a mean decrease of 29 %, but for individual stations, such as LDB0 and LDB2, it could reach up to 50 %. This can be also understood as the impact of removing systematic errors from the residuals or, in other words, as a degradation of STDs quality when applying uncleaned

residuals due to the contamination by these systematic errors. While estimating STDs, it should thus not be recommended to add raw residuals. However, this comparison does not suggest any preference for either using purely the estimated model (no residual) or adding clean residuals to STDs. Both approaches still comprise of errors due to approximations, local



environmental effects, instrumentation effects or processing models. Additionally, the impact of cleaning the post-fit residuals for the reconstruction of STDs can be characterized by a median SDEV of 2.6 mm projected into the zenith direction when calculated from differences between raw and clean STDs over all solutions and stations, see the last two data columns in the table.

**Table 3**: Statistics from comparisons of individual GNSS STDs (projected in the zenith direction) while using none, raw and clean residuals; median values of biases and standard deviations (SDEV) calculated over all stations with an exception of LDB0 station are given.

| Solution | nonRES – rawRES | | nonRES – clnRES | | rawRES – clnRES | |
|---|---|---|---|---|---|---|
| | Bias [mm] | SDEV [mm] | Bias [mm] | SDEV [mm] | Bias [mm] | SDEV [mm] |
| GFZ | +0.02 ± 0.03 | 3.88 ± 0.51 | -0.01 ± 0.01 | 2.77 ± 0.19 | -0.01 ± 0.03 | 2.73 ± 0.67 |
| GOP_F | -0.00 ± 0.02 | 4.69 ± 0.41 | +0.00 ± 0.01 | 3.43 ± 0.19 | -0.01 ± 0.01 | 3.14 ± 0.50 |
| GOP_S | -0.00 ± 0.01 | 4.39 ± 0.42 | -0.01 ± 0.00 | 3.12 ± 0.16 | -0.01 ± 0.01 | 2.99 ± 0.53 |
| ROB_G | +0.02 ± 0.05 | 3.59 ± 0.66 | +0.02 ± 0.02 | 2.66 ± 0.30 | +0.00 ± 0.04 | 2.37 ± 0.67 |
| ROB_V | +0.01 ± 0.05 | 3.58 ± 0.67 | +0.02 ± 0.02 | 2.66 ± 0.30 | +0.01 ± 0.04 | 2.37 ± 0.67 |
| TUW_3 | +0.03 ± 0.06 | 3.90 ± 0.75 | -0.01 ± 0.04 | 2.85 ± 0.35 | -0.02 ± 0.06 | 2.63 ± 0.78 |
| TUW_7 | +0.04 ± 0.05 | 3.89 ± 0.75 | -0.01 ± 0.04 | 2.80 ± 0.35 | -0.02 ± 0.04 | 2.60 ± 0.78 |
| WUE | +0.02 ± 0.04 | 3.64 ± 0.49 | +0.00 ± 0.01 | 2.54 ± 0.19 | -0.02 ± 0.04 | 2.50 ± 0.66 |

Individual GNSS solutions provided also variants using the same software and strategy, but with modified settings. This allows us to assess its impact on the estimated parameters, see Table 4. Consequently, we evaluated STDs calculated without residuals expecting the impact (mainly) on estimated ZTDs and horizontal gradients. Biases reached a sub-millimetre level and were almost insignificant with a single exception of using GMF versus VMF1 mapping function resulting in a positive bias of +1.2 mm over all stations. Studied effects were sorted then by the magnitude of their standard deviations.

Surprisingly, the impact of the elevation angle cut-off (3° versus 7°) resulted in a minimum mean standard deviation below 1 mm, see TUW_3 and TUW_7. The use of mapping functions based on climatology (GMF) or meteorological (VMF1) data resulted in a larger impact, at the level of 2 mm, which is similar as the impact found for using single (GPS) or dual (GPS+GLO) GNSS constellations. The use of different temporal resolutions of ZTDs and gradients could not be avoided among various contributions due to limited capabilities of handling high number of parameters. An assessment of the

temporal resolution will be also influenced by applying relative constrains in deterministic approach or setting a noise level in stochastic process. Anyway, we compared two solutions (ROB_V and TUO_R) using the Bernese software and DD method with the same settings, but different temporal resolutions of ZTDs and gradients. The results show discrepancies at a level of 3 mm which could be partly explained by different sampling. However, we assume also contributions from specific differences in strategies such as data pre-processing. Last but not least, the impact of using Kalman filter for simulating real-

time solution compared to the back-smoother (offline) solutions resulted in the lowest agreement of 4.8 mm in terms of the standard deviation calculated from differences.



**Table 4**: Impact of selected strategy modifications assessed via comparing individual STDs solution variants. Median values of biases and standard deviations (SDEV) calculated over all stations with an exception of LDB0 station using the estimated model only (without residuals) are given.

| Compared solutions | Remarks on solution differences | | Bias [mm] | SDEV [mm] |
|---|---|---|---|---|
| TUW_3 – TUW_7 | *Elevation angle cut-off:* | 3° versus 7° | +0.46 ± 0.69 | 0.98 ± 0.45 |
| ROB_G – ROB_V | *Mapping function:* | GMF versus VMF1 | +1.20 ± 0.20 | 1.91 ± 0.27 |
| TUO_G – TUO_R | *GNSS observations:* | GPS versus GPS+GLO | +0.66 ± 0.37 | 2.01 ± 0.47 |
| ROB_V – TUO_R | *ZTD/gradient resolution:* | 15min/1h versus 1h/3h | -0.19 ± 0.34 | 3.10 ± 0.40 |
| GOP_F – GOP_S | *Processing strategy:* | Kalman filter versus backward smoothing | -0.60 ± 0.55 | 4.81 ± 0.79 |

Figure 7 provides an evaluation of the STDs at their original elevation angles for the station POTS. Four individual panels show bias (top left), normalized bias (NBIAS, top right), standard deviation (bottom left), and normalized standard deviation (NSDEV, bottom right). Normalized bias and normalized standard deviation were computed to see the dependence of relative errors in STDs at different elevations. For its computation, absolute differences of STDs from two solutions were divided by the STD values from the reference solution. For example, when the solution from GFZ (taken here as the reference) was compared against TUO, the standard deviation was computed from all valid absolute differences given as

$$diff\_absolute \ = \ STD_{GFZ}^{i} - STD_{TUO}^{i} \tag{16}$$

and normalized standard deviation from all valid relative differences given as

$$diff\_relative \ = \ (STD_{GFZ}^{i} - STD_{TUO}^{i})/STD_{GFZ}^{i} \tag{17}$$

Since STDs are reconstructed mainly from ZTDs and horizontal gradients, any small differences between the two solutions in the zenith direction should become much larger after mapping down to lower elevations. Therefore, higher values of bias and standard deviation are expected with the decreasing of elevation angle. Indeed, we found that the agreement among individual solutions compared to the GFZ STDs is rather stable above the elevation angle of 30 degrees. Corresponding biases of individual elevation bins are almost constantly within ±4 mm and standard deviations are slowly increasing up to 10 mm at 30 degrees. However, both increases significantly for bins below 30 degrees which is mainly pronounced for standard deviations following an exponential decay up to 50 mm at 7 degrees. Biases are also more dependent on solutions and stations, and for example it strongly deviates in the case of ROB_V and ROB_G solutions from the others. Both perform contrary to each other below 30 degrees in particular case of the station POTS while both use the Bernese GNSS Software and the same strategy, but different mapping function only. ROB_V reached mean offsets up to +17 mm for the lowest elevation angles while ROB_G gave mean negative offsets down to -12 mm. Note also that ROB_V is consistent with TUO_G. Since STDs from ROB_V at higher elevations bins are also negative compared to GFZ, it compensated the extreme behaviour at the lowest elevation angles and the overall offset in the zenith direction becomes negative too. The difference between ROB_G and ROB_V biases at low elevations are thus not so apparent in the zenith statistics, but still visible at the level of 1 mm (Figure 4). The positive bias of GMF versus VMF1 - visible in Table 4 - can be explained as the effect



prevailing in the modelling observations at the elevation angles below 30 degrees. Similarly, overall biases in the ROB_V solution compared to ROB_G are found at low elevation angles for all other stations though the situation is not always as significant as for the station POTS. It should be also noted that the conclusion is independent of adding or not any residuals.

In terms of standard deviation, the presumption about the dependency of statistics on the elevation angle is clearly visible in

the increasing errors with the decreasing elevation angles (Figure 7). Normalized standard deviation remains almost constant over all elevation angles indicating a very consistent relative performance of STDs among all the solutions. A similar behaviour is present at all stations although the absolute values can be higher for some stations or solutions, namely GOP_F for LDB0 and WTZZ with standard deviations reaching up to 72 mm.

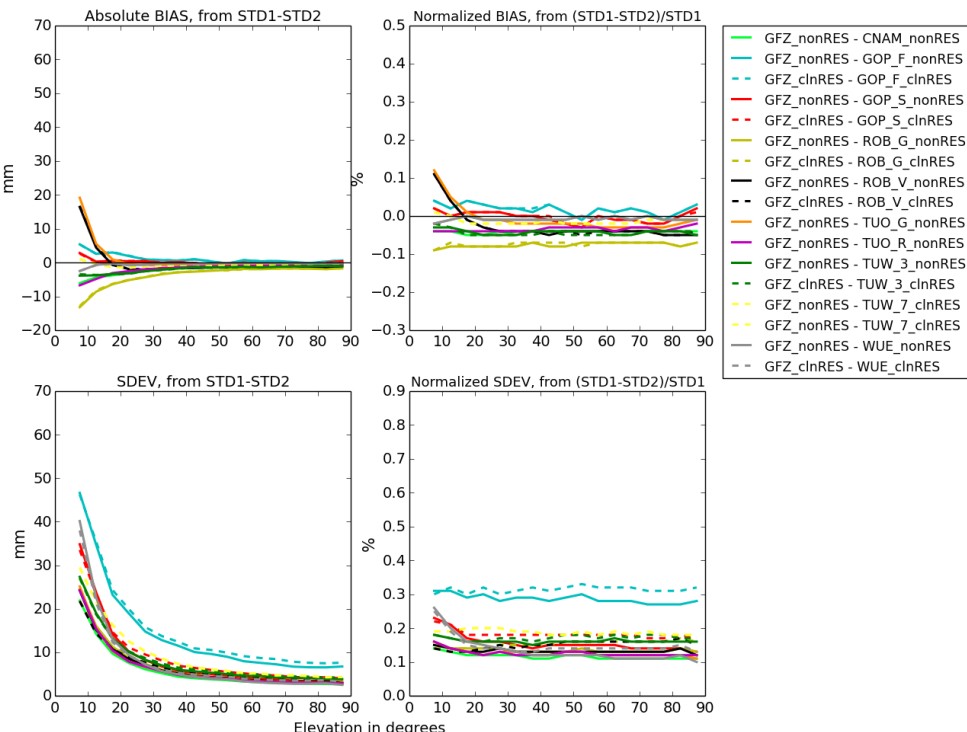

Figure 7: Comparison of individual GNSS STD solutions against GFZ STD solution at station POTS, in slant directions.

## 7.2   GNSS versus NWM

STDs from four individual NWM ray-tracing solutions delivered by three different institutions entered the validation. Only NWM-based results given at 00, 06, 12, and 18 UTC were used. To ensure the consistency of the comparison, only epochs for which STD values were available in all GNSS solutions were considered, i.e. if a single STD value was missing in any

GNSS solution, then the STD values at the same epoch were also removed from all other GNSS solutions. This selection of observations and the low time resolution of the NWM models (six hours) led to a restricted set of STDs available for the validation consisting of 9,866 of observations in total.



### 7.2.1 Evaluation of all GNSS solutions without residuals in the zenith direction

Figure 8 presents the comparison of individual NWM STDs and GNSS STDs (without residuals) expressed in the zenith direction. From top to bottom, plots show biases (left) and standard deviations (right) for ALA/BIRA, ERA/GFZ, GFS/GFZ and ALA/WUELS. For most stations, the bias varies between -5 mm and +3 mm for the ALA/BIRA solution, with all GNSS solutions performing similarly. Slightly higher biases and more variability between GNSS solutions are observed at the station POTM. This behaviour is to account at the side of the GNSS solutions, since POTM and POTS are collocated and the ALA/BIRA provide the same STDs for the validation at both stations. If we exclude both GOP solutions and the GFZ solution the range of biases at station POTM is very similar to range at station POTS. The difference in height of those two stations is 0.5 m. The station POTS is equipped with a choke ring antenna while the station POTM is not. This could cause the slightly higher range of biases for individual solutions which occurred at station POTM. Significant biases of approximately -20 mm are present at two Austrian stations, KIBG and SAAL, and being similar for all GNSS solutions. Both stations are situated in the mountainous area south-west of Salzburg city. Since the same biases do not occur at GNSS versus ERA/GFZ nor GNSS versus GFS/GZF comparisons, they are most likely due to a deficiency of the ALADIN-CZ orography representation. Note that ALA/BIRA and ALA/WUELS STDs show an unexpected opposite behaviour for KIBG and SAAL stations (Figure 8), which is related to the difference in the strategy used. This is possibly due to the estimation of the altitude of parameters, their interpolations, and the difference in the step of integration. Except at those two stations, similar biases as for ALA/BIRA can be also found for the GNSS versus ERA/GFZ comparison, ranging from -3 mm to +7 mm (+11 mm at POTM). Although the bias characteristics for GFS/GFZ are practically identical to those obtained for ERA/GFZ, the results for the NCEP GFS model are shifted by approximately +5 mm, resulting into biases ranging from +3 mm to +12 mm (+17 mm at POTM). The origin of this systematic deviation was identified in ZWD values estimated from the GFS model (Douša et al., 2016), and understood as the effect of the lower vertical resolution of NCEP GFS model compared to other NWMs, leading to larger errors in vertical interpolations.

Standard deviations between GNSS STDs and ALA/BIRA, ERA/GFZ and GFS/GFZ solutions are usually around 10 mm. Generally, they are higher than the comparison of individual GNSS solutions presented in Section 7.1 and they are also more station dependent. Degradations can be observed at mountainous stations KIBG and SALL for the ERA/GFZ, GFS/GFZ, and ALA/BIRA STDs, reaching standard deviations up to 18 mm in case of the ERA-Interim NWM.

ALA/WUELS solution performed differently compared to all other NWM solutions. It is biased against GNSS solutions, with biases ranging from +9 mm to +25 mm and highest values observed at stations KIBG and SAAL. Standard deviation values are also much higher, by about a factor of 2.5 worse compared to values obtained from the GNSS versus GFS/GFZ comparison. The probable reason for this is that signal tracking was performed for vacuum elevation angles. The impact is especially visible at low elevation ray-paths at which the signal has to travel through the troposphere for a longer time, enhancing the negative effect of underestimated delays.





Figure 8: Comparison of individual GNSS STD solutions without residuals (nonRES) against NWM solutions ALA/BIRA, ERA/GFZ, GFS/GFZ, ALA/WUELS (from top to bottom), projected in the zenith direction: bias (left) and standard deviation (right).





Evaluation of the influence of raw and cleaned post-fit residuals on slant total delaysChyba! Nenalezen zdroj odkazů.Comparisons between the three versions of GNSS solutions (nonRES, clnRES, rawRES) and the ALA/BIRA, ERA/GFZ, and GFS/GFZ NWM solutions were done to test the influence of post-fit residuals on GNSS STDs. The ALA/WUELS solution was excluded from this comparison because of the lower quality of its STDs. All GNSS solutions without post-fit residuals reached slightly lower standard deviation values than the solutions which included either raw or cleaned post-fit residuals, while differences in biases were negligible. An average increase of standard deviation was 4.5 % for clean residuals and 8.3 % for raw residuals. Indeed, because of their low horizontal and time resolution, the used NWP models can barely capture the very fine-scale tropospheric structures which are supposed to be included in the GNSS residuals. As a consequence, this comparison does not allow to conclude clearly on the potential benefits of cleaned post-fit residuals in the reconstruction of the GNSS STDs.

### 7.2.2 Evaluation in the slant direction

Statistics from the comparison of ALA/BIRA. ERA/GFZ and GFS/GFZ against GNSS solutions expressed at original elevation angles of slant delays are presented for the station POTS in Figure 9. Full lines display the median over all GNSS solutions and dashed lines represent minimum and maximum ranges. Significantly higher biases can be found at the lowest elevation bin in all three solutions and at all stations (not displayed). At some stations, sudden increases of bias at individual elevation bins were observed. They happen at any kind of elevation angle (different for each 3 comparisons) and are particularly visible in terms of normalized bias. These sudden increases of the bias might found its origin either in the fact that the model sometimes cannot render the tropospheric structures at their exact locations (unexpected location of high/low values of water vapour partial pressure), or because models running at these resolutions have a tendency to smooth out such tropospheric heterogeneities. Comparing with a model running at convective-permitting scale (e.g. 1 to 4 km) would help to sort out if the origin of such behaviours is to account on the NWP model STD side or on the GNSS STD side.

For all stations, standard deviations present the shape with significantly higher values at elevations below thirty degrees followed by a gentle decrease towards the zenith direction. An exception was found at stations WTZR, WTZS and WTZZ where a rather smooth shape of the curve is disrupted with sudden increases and decreases of standard deviation at particular bins over all elevation angles. This implies mainly for GNSS versus ALA/BIRA solution. GNSS versus ERA/GFZ and GNSS versus GFS/GFZ results show such increases and decreases low frequently and with a lower magnitude by a factor of two or three. Normalized standard deviations vary at all elevation angles for all validated stations. The values range between 0.2 % and 0.9 % with the highest values occurring at high elevation angles.

Results from the GNSS versus ALA/WUELS solutions (not presented) show enormous increase of both absolute and normalized bias and standard deviations at low elevation angles below 25 degrees at all stations. It even reached biases up to 350 mm and standard deviations up to 300 mm at some stations. Statistical parameters became more stable above 25 degrees, with occasional disturbances similar to those observed in other NWM-based solutions.



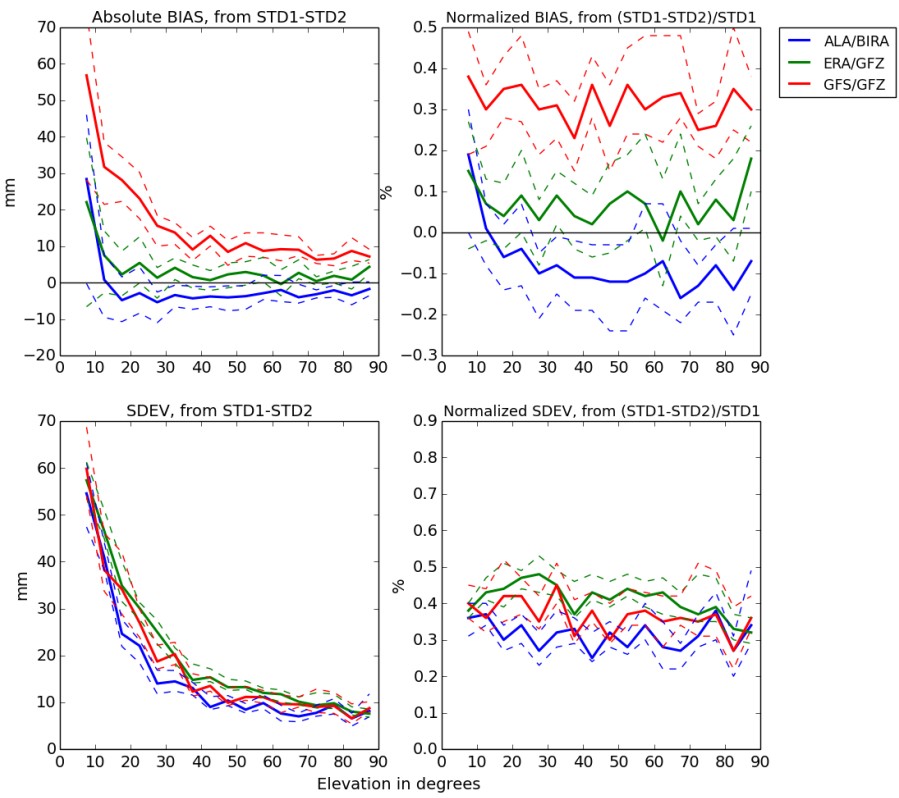

Figure 9: Comparison of NWM-based solutions (ALA/BIRA, ERA/GFZ and GFS/GFZ) against GNSS solutions at station POTS, in the slant direction; full line represents a median of all GNSS solutions, dashed lines show minimum/maximum range for GNSS solutions.

**7.2.3     Summary of results for GNSS versus NWM**

A summary of the GNSS versus NWM validation is presented in Table 5. For each reference station a median of bias and a median of standard deviation in the zenith direction between all GNSS solutions and a particular NWM-based solution are given. If we consider ALA/BIRA and ERA/GFZ only, without the two mountainous stations KIBG and SAAL, absolute biases between NWM and GNSS solutions stay mostly below 3 mm, which represents a very good agreement between these

independent sources used for retrieving slant delays. Standard deviations generally range from 8 mm to 12 mm, with the exception of ALA/WUELS showing lower precision by a factor of 2.5.

In this paper, statistics stems from the complete benchmark period, and it should be noted that the stability on a daily time scale was much better for GNSS STDs than for NWM ray-traced STDs. Significantly higher values of biases and standard deviations were observed at particular days for NWM solutions. A detail evaluation of daily statistics with a respect to the

extreme weather conditions is one of the topics that we will study in future.



**Table 5:** Medians of bias and standard deviation values of differences between all GNSS solutions and a particular NWM-based solution at each reference station, expressed in the zenith direction.

| Station | Bias (mm) | | | | Standard deviation (mm) | | | |
|---------|-----------|--------|--------|-----------|-------------------------|--------|--------|-----------|
| | ALA/BIRA | ERA/GFZ | GFS/GFZ | ALA/WUELS | ALA/BIRA | ERA/GFZ | GFS/GFZ | ALA/WUELS |
| GOPE | 0.0 | 2.9 | 8.3 | 11.1 | 8.4 | 10.3 | 7.2 | 22.4 |
| KIBG | -19.2 | 5.0 | 9.7 | 22.6 | 11.7 | 17.7 | 11.0 | 26.7 |
| LDB0 | -1.1 | 1.6 | 6.4 | 11.5 | 10.0 | 10.4 | 8.9 | 26.2 |
| LDB2 | -1.5 | 1.0 | 6.2 | 15.1 | 9.2 | 10.1 | 8.7 | 25.4 |
| POTM | 2.8 | 5.7 | 12.0 | 18.4 | 8.0 | 10.6 | 9.4 | 26.2 |
| POTS | -1.9 | 1.3 | 7.4 | 12.4 | 7.7 | 10.3 | 9.2 | 25.8 |
| SAAL | -19.9 | 7.3 | 11.1 | 23.8 | 12.6 | 17.7 | 11.7 | 22.9 |
| WTZR | -4.6 | -1.5 | 4.9 | 10.4 | 10.8 | 11.7 | 8.5 | 22.9 |
| WTZS | -3.1 | -0.5 | 4.7 | 11.2 | 11.4 | 12.1 | 8.5 | 23.6 |
| WTZZ | -2.4 | 0.9 | 5.9 | 11.6 | 11.4 | 12.1 | 9.1 | 23.7 |

### 7.3    GNSS versus WVR

Figure 10 compares GNSS and WVR solutions at stations POTM and POTS, in the zenith direction. The number of slant observations which entered the comparison was 32,794 at station POTM and 36,070 at station POTS. Two remarks can be done on the evaluation of biases. Firstly, an overall bias of about 4 mm between the stations POTM and POTS, identified for all GNSS solutions in Figure 8, indicates a common issue with the GNSS data processing at the station POTM, particularly diverging for GOP_F, GOP_S and GFZ solutions. Secondly, a bias of about 5.5 mm in the zenith direction can be found between WVR and GNSS solutions at station POTS. This bias roughly corresponds to 1 kg/m$^2$ of Integrated Water Vapour (IWV), what can be addressed as the achievable accuracy of any technique.

Values of standard deviation, resulting mostly in 12 mm, are higher than what was observed in any GNSS versus GNSS comparisons (Section 7.1). It is also slightly higher than the GNSS versus NWM comparisons (Section 7.2). A cut-off elevation angle of 15 degrees was used for the comparison of GNSS versus WVR STDs, in contrast to the validations of other sources (7 °). Consequently, the largest differences in STD values found at very low elevation angles did not enter this validation. Additionally, it has to be noted that the results can be partly influenced with the settings applied for finding pairs between GNSS and WVR STDs (Section 6). STDs from WVR can thus originate from slightly different azimuth/elevation angles and times than the GNSS ones. All GNSS solutions perform very similarly against WVR, except the GOP_F as expected due to the real-time methodology applied. Generally, standard deviations for all solutions using cleaned residuals (resp. raw residuals) are in average 1.7 % (resp. 3.8 %) higher than for the solutions without residuals. Although the differences between solutions variants are smaller, the situation is in a good agreement with the results obtained for GNSS versus NWM comparisons presented in Section 7.1. Cleaning of post-fit residuals proved to be valuable, however, the difference between versions of solutions with no residuals and cleaned residuals are too small to allow a decision if residuals




should be avoided at all or if the cleaning filter still needs improvements to better benefits the STD reconstruction. Therefore, more investigation around the post-fit residuals cleaning needs to be done.

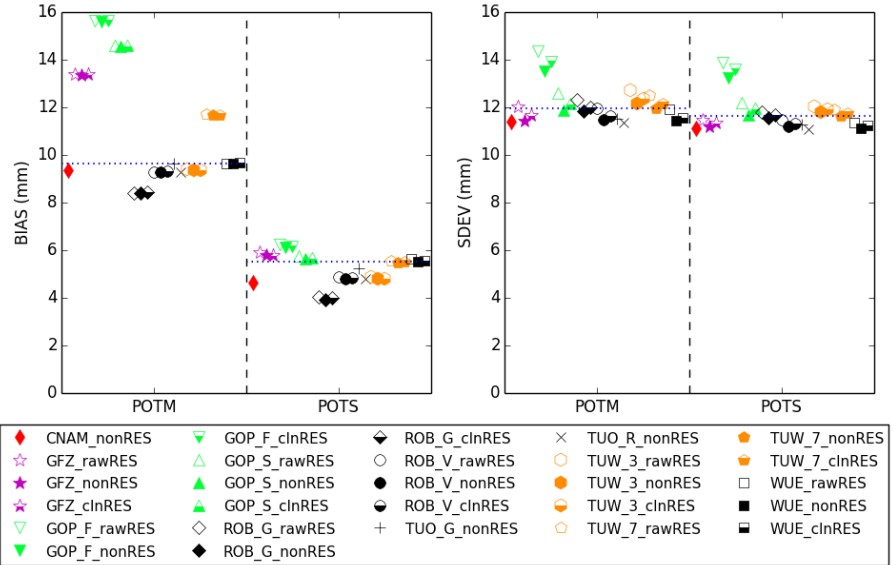

Figure 10: Comparison of individual GNSS STD solutions for stations POTM and POTS versus WVR measurements, expressed in the zenith direction, bias (left) and standard deviation (right). The median value of all solutions at each station is represented by the dotted blue line in each bin.

The GNSS versus WVR validation at the station POTS using original elevation angles is displayed in Figure 11. The decrease of values of four statistical parameters strongly follows the increase of elevation angle. Although some differences between solutions are visible, all of them performed very similarly. High values of the normalized standard deviation at lower elevation angles indicate difficulties for the WVR to provide high-quality observations at low elevations. A sudden increase of the values is visible at elevations between 55–60 degrees most likely originating from WVR observations which has not been understood yet.





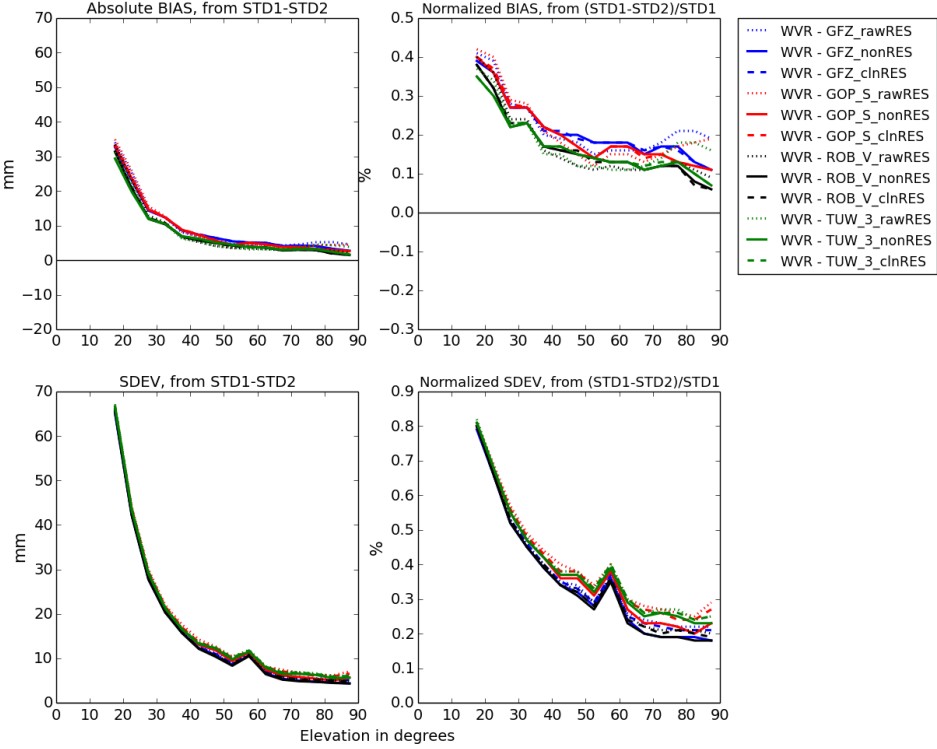

Figure 11: Comparison of WVR against individual GNSS STD solutions at station POTS, in the slant direction.

## 8 Results at dual stations

Dual stations were available in the benchmark campaign at three different locations in Germany. The first two sites collocate twin GNSS reference stations (LDB0+LDB2 and POTM+POTS), the third location collocate three individual reference stations (WTZR+WTZS+WTZZ). Nevertheless, in the case of Wettzell, only results for WTZR+WTZS are presented due to their similarity with the two other combinations at the same place. Characteristics of the stations are summarized in Table 6. This comparison aims at validating the (internal) accuracy of STDs based on the presumption that STDs from collocated receivers should be very similar from the atmospheric point of view. Results are presented hereafter for each location.



**Table 6**: Characteristics of individual dual stations.

| Dual station | Location | Horizontal distance (m) | Vertical distance (m) | Identical type of receiver | Identical type of antenna | Pairs of observations |
|---|---|---|---|---|---|---|
| LDB0+LDB2 | Lindenberg | 177 | 0.6 | NO | NO | 143,005 |
| POTM+POTS | Potsdam | 2.5 | -0.5 | NO | NO | 180,636 |
| WTZR+WTZS | Wettzell | 69 | 2.6 | NO | YES | 84,443 |

Two sets of STDs from the same solution, but different GNSS reference station, were compared to each other. Table 7 and Table **8** show the statistics expressed in the zenith direction for observations at ranging elevation angles from 7 to 15 and from 15 to 90 degrees, respectively. The biases stay very stable regardless if post-fit residuals are used or not. The lowest values of standard deviation for all dual stations can be found for the solutions without using post-fit residuals (nonRES), indicating that the residuals are still strongly site-specific, i.e. not only representing the effect due to the local asymmetry in the water vapour distribution around the GNSS station. Additionally, by cleaning the systematic portion in residuals, we are not able to remove all instrumentation, multipath or other local effects sufficiently. Anyway, using raw post-fit residuals from GNSS analysis without additional cleaning should always be avoided. Interestingly, comparing the statistics for STDs evaluated from 7-15 and 15-90 degree ranges, the standard deviations are smaller in high elevations for variant not using residuals, while these are higher for low elevations when using either cleaned or raw residuals. This can be interpreted as the standard GNSS tropospheric model (ZTD and horizontal gradients) represents very well observations at elevations above 15 degrees, but suffers by the modelling deficiencies mainly at low elevations. The negative effect of adding cleaned or raw residuals is then more pronounced in the statistics for STDs at high elevations.



**Table 7**: Comparison of GNSS STDs from the elevation angles ranging from 7 to 15 degrees at three dual stations; median values of biases and standard deviations (SDEV) calculated over all GNSS STD solutions are given; statistics are expressed in the zenith direction.

|  | nonRES | | clnRES | | rawRES | |
|---|---|---|---|---|---|---|
|  | Bias [mm] | SDEV [mm] | Bias [mm] | SDEV [mm] | Bias [mm] | SDEV [mm] |
| LDB0+LDB2 | -1.60 | 3.47 | -1.58 | 4.43 | -1.57 | 5.35 |
| POTM+POTS | -6.00 | 1.91 | -5.98 | 3.20 | -6.67 | 4.00 |
| WTZR+WTZS | -0.11 | 2.13 | -0.09 | 3.34 | -0.05 | 3.87 |

**Table 8**: Comparison of GNSS STDs from the elevation angles ranging from 15 to 90 degrees at three dual stations; median values of biases and standard deviations (SDEV) calculated over all GNSS STD solutions are given; statistics are expressed in the zenith direction.

|  | nonRES | | clnRES | | rawRES | |
|---|---|---|---|---|---|---|
|  | Bias [mm] | SDEV [mm] | Bias [mm] | SDEV [mm] | Bias [mm] | SDEV [mm] |
| LDB0+LDB2 | -0.56 | 3.11 | -0.55 | 5.17 | -0.46 | 7.97 |
| POTM+POTS | -6.14 | 1.68 | -6.13 | 3.26 | -6.08 | 4.54 |
| WTZR+WTZS | -0.22 | 1.96 | -0.21 | 3.93 | -0.22 | 5.17 |

10   Figure 12 displays results for comparison of individual dual stations in the slant direction. Strong variations are observed mainly in normalized biases over all elevation angles for the solutions using raw post-fit residuals (rawRES). These are clearly related to local effects such as multipath or instrumentation. The systematic variation in STD differences at dual stations disappears after the residuals cleaning as obvious from the solutions using cleaned residuals (clnRES). The standard deviations as well as normalized standard deviations at all stations are clearly the lowest for variants not using post-fit
15   residuals (nonRES), a little worse when using cleaned residuals, and by much worse when using raw residuals.



Figure 12: Comparison of GNSS STDs at dual stations from individual GNSS solutions in the slant direction, dual stations from left to right: LDB0-LDB2, POTM-POTS, WTZR-WTZS. Statistical parameters from top to bottom: bias, normalized bias, standard deviation, normalized standard deviation.

## 9    Conclusions

In this paper, we presented results of validating tropospheric slant total delays obtained from GNSS data processing with those obtained from NWM ray-tracing, WVR measurements and collocated GNSS stations, focusing on the optimal method for estimating GNSS STDs. Ten GNSS reference stations were selected, exploiting data from a 56-day benchmark period. Eleven GNSS solutions, four NWM-based solutions and one WVR-based dataset entered this validation study. Eight out of eleven GNSS solutions delivered STDs in three variants: 1) without post-fit residuals, 2) with raw post-fit residuals, and 3)



with cleaned post-fit residuals. The comparisons were carried out into two scenarios, firstly for STDs at their true elevation angles, and secondly, for STDs mapped to the zenith direction using a common simple mapping function of 1/sin(e), e being the elevation angle.

All GNSS solutions without residuals were compared against the GNSS solution without residuals provided by GFZ Potsdam, which was selected as the reference. Almost all solutions were in a very good mutual agreement although many different software, strategies and settings were used. Absolute biases between GFZ and other solutions were within ±3 mm for all individual stations and standard deviations were ranging from 3 mm to 5 mm in the zenith direction. An exception was the GOP_F solution - designed for a real-time demonstration capability - with standard deviations around 7 mm. Comparison of variants of individual STD solutions without residuals, with raw and with cleaned residuals was used to study the impact of different strategies for optimally retrieving STDs from GNSS (i.e. including a maximum of relevant information relative to the asymmetry of the local troposphere). The impact of adding cleaned residuals reached 2.5 - 4.5 mm in the zenith direction, while using raw residuals instead resulted in increased discrepancies at the level of 3.0 - 6.5 mm, with a pronounced station dependency. The impact was practically negligible in terms of systematic errors remaining around ±0.1 mm for raw residuals and less than ±0.1 mm for cleaned residuals, thus fully negligible.

GNSS STDs were then validated against STDs obtained from NWM ray-tracing. Bias and standard deviation values between GNSS and NWM solutions strongly depended on ray-tracing method, NWM source and individual station. Significantly worse results, by a factor of 2.5 in terms of standard deviation, was shown for the ALA/WUELS solution. The origin was identified as mainly a deficiency in the ray-tracing methodology. Biases in the zenith direction remained usually below ±3 mm for ALA/BIRA and ERA/GFZ solutions, while a positive bias of around 6 mm was observed for NWM_GFS solution. Standard deviations for all GNSS versus NWM comparisons were similar with a small oscillation around 10 mm, when excluding the ALA/WUELS solution. Normalized standard deviation values did not remain stable throughout all elevations as in the case of GNSS versus GNSS comparisons. Their values varied at all elevation angles and over stations reaching often the lowest values at elevation angles below 15 degrees.

GNSS STD solutions from stations POTM and POTS were validated against collocated WVR observations. A positive bias of around 5.5 mm was found for the WVR instrument when it was compared to GNSS STDs from POTS station. Standard deviations for GNSS versus WVR comparisons reached about 12 mm in the zenith direction, and were higher than the compare with NWM solutions. GNSS STDs without post-fit residuals agreed slightly better than their versions including either raw or cleaned residuals.

STDs from collocated GNSS reference stations using the same solution were confronted in order to validate the impact of post-fit residuals. For this purpose, GNSS stations at three different locations were evaluated. STDs from all solutions without including post-fit residuals reached always the lowest standard deviations compared to the solutions with post-fit residuals. We found a strong elevation dependency of bias for the variant using raw residuals and the discrepancies were observed generally larger for higher than smaller elevation angles. This strong elevation dependency almost vanished for the variant using cleaned residuals.



Based on this validation study, we do not recommend adding raw post-fit residuals into STDs since residuals still contain systematic effects which surpass the tropospheric information content. As already mentioned in Bender et al. (2008) and Kačmařík et al. (2012), cleaning post-fit residuals improves the situation considerably. However, similarly to what was found by Kačmařík et al. (2012), variants of GNSS STD solutions generated without post-fit residuals still reached better

agreement with NWM and WVR solutions than those with cleaned residuals indicating probably that we are currently not able to remove completely all other effects due to the local troposphere. The use of clean residuals for STD retrievals could be therefore recommended only after improving the strategy to screen and remove all undesirable effects what is a matter of a further study. On the other hand, most of the GNSS solutions used one hour or less for the estimating tropospheric horizontal gradients, which assumes that the first order asymmetry in the troposphere can be partly captured by them,

although often averaged by potentially low temporal resolution compared to their real dynamics.

It has to be also noted that used WVR and NWM STDs have their own limitations and part of the difference originates just from the fact that different techniques sense by essence differently the local asymmetry of the troposphere. Unless the NWM model is run at a convection-permitting scale with proper physic models inside and with a quick update cycle and the WVR is correctly calibrated and delivers observations exactly in the direction of GNSS satellites at desired epochs, STDs from

those two techniques can be hardly as reliable as is needed.

Last but not least, we should discuss an additional complexity of the GNSS processing when using cleaned post-fit residuals to reconstruct the STDs. In addition to the standard GNSS data processing, i.e. the estimation of ZTD/ZWD and horizontal gradient parameters, information about azimuth/elevation angles for all observations and corresponding residuals need to be stored along with all solutions. The cleaning of systematic effects (e.g. multipath, environmental effects and antenna phase

centre variations) requires statistical information about the residuals over tens of days in order to provide reliable stacking maps characterizing them properly. All that lead to additional complexities and increased computation load when targeting operational provision of STDs (including cleaned residuals) for weather forecasting.

### Acknowledgement

This study has been organised within the E.U. COST Action ES1206 (GNSS4SWEC). The authors thank all the institutions

that provided data for the benchmark campaign (Douša et al., 2016) on which the validation was based on. Namely we want to thank S. Heise (GFZ) for providing the WVR data. The GFS data were provided by the National Centers for Environmental Prediction (www.ncep.noaa.gov). The ERA-interim data were provided by the European Centre for Medium-Range Weather Forecasts (http://www.ecmwf.int/en/forecasts/datasets). M. Kačmařík, J. Douša and P. Václavovic acknowledge the support from the Czech Ministry of Education, Youth and Sports (project no. LD14102). E. Pottiaux

(ROB) and H. Brenot (BIRA) acknowledge the support from the Solar-Terrestrial Centre of Excellence (STCE). J. Kapłon and P. Hordyniec (WUELS) acknowledge the support of Polish National Science Centre (project no. UMO-



2013/11/D/ST10/03473) for financial support and Wroclaw Center of Networking and Supercomputing (http://www.wcss.wroc.pl) for computational grant using Matlab Software License No: 101979.

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
