# Peer review of "Inter-technique validation of tropospheric slant total delays"

_Atmospheric Measurement Techniques, 2016_

## Referee Comment (RC1) · Anonymous Referee #2 · 26 Jan 2017

This manuscript describes inter-comparisons of slant total delays (STDs) derived from GNSS solutions, numerical weather models (NWM), and radiometers (WVR). In comparisons between GNSS software, the authors found most of them show good agreements with each other. Moreover, they recommend no use of raw post-fit residuals whereas STDs without residuals and with cleaned residuals are better. As for NWM and WVR, the authors concluded they are not reliable due to their large errors. I roughly agree with their conclusions. However, since I have to point out some weaknesses in this manuscript, it should be published after major revisions.

Major comments 1) Residuals STD is decomposed with ZTD, gradients (G) and residuals. Since Equation 1 only represents the first two terms, the residual term should be added.

[Figure]

Since the horizontal scales of ZTD, G, and residuals are 500, 50, and 5 km (Shoji et al. 2004), it is important to add residuals into STDs when convective activities are considered in any studies using STDs like this manuscript. From the view point of this, the authors have no chance to avoid residuals in processing STDs. In addition, residuals should be cleaned as pointed by Shoji et al. (2004). Therefore, I don't agree that "the usage of the information content from the post-fit residuals for the reconstruction of the STDs remains an open question" in this study (L5 P5). The authors should re-consider the effect of post-fit residuals in their formulation and re-organize this manuscript from the view point that they really need to investigate the effect of post-fit residuals in this study.

2) Comparison in the zenith direction The authors compared STDs in the zenith direction using mapping functions. Sine these functions were made statistically (excluding gradient on the day), I recommend the authors to use STDs only in high elevation angles (> 60 or 70 degree) for the comparisons. This is especially useful in comparison of GNSS vs WVR, because it is able to avoid errors of surface pressure gradient in calculating STDs from WVR.

3) Comparison with NWM There are three models appeared in this manuscript: ERA-Interim, NCEP GFS, and ALADIN-CZ. To help the readers understand the discussion on this topic, please describe more settings on these models.

- Within these, only ALADIN is a regional model, and others are global. State their general characteristics more. - Do all of these models assimilate GNSS data for their initial conditions or not? - How large are their grid spacings? - ERA-Interim is the ECMWF re-analysis data produced 6 hourly. I guess GFS is 6 hourly operational analysis data at NCEP. What is ALADIN-CZ? - Does ALADIN-CZ have large domain enough to produce STDs? I concern that STDs at low elevation angles might penetrate the lateral boundary of the model and need special treatment like STDs over the top of the model. - ALADIN-CZ may be a cloud-permitting model (this depends on its resolution) with explicit cloud microphysics. In this case, it is possible to calculate STDs with hydrometer
effect. Is this right? If yes, I suggest to do this (see the major comment 5). P7 section 4 There are two error sources in this comparison; STD solutions and NWMs. I suggest the authors to employ single STD solution with three NWMs and then compare the results with observed STDs. This makes error sources reduced single (only NWM) and discussion much easier (section 7.2).

4) Figures and discussion Although there are many graphs appeared in this manuscript, some of them are not appropriate for discussion. For instance, though Figure 12 displays 30 lines in 12 panels, the authors made a discussion only in single paragraph (P30 L10). Another example. Although the authors showed small number of figures in connection with GNSS versus NWM comparison, they discussed many points (stations) without figures in section 7.2. I recommend to re-organize discussion and figures.

5) Assessment of components in the atmosphere Although the discussion on the effect of each component of the atmosphere (section 4.4) is important, the authors did not show any conclusion. I suggest to examine the same effect using NWMs additionally and illustrate useful information.

Minor comments P1 L21: "between GNSS a NWM" Reword to "between GNSS and NWM".

P1 L29: "along his path" Reword to "along the path"

P4 L8: "was operating only" Reword to "was operated only" (?)

P6 L91: "three variants of the solution" I don't think that it is worth to examine "nonRES" case in this paper. See my major comment.

P8 L4: "mix ratio of liquid" mixing ratio of liquid

P10 L24: "The contribution of water – neglected in the total delay." As I mentioned in my major comments, I suggest the author to examine these contributions.

P11 Figure 1 Enlarge the land names.

P12 L4: "Figure 2 shows simulated STDs" P12 L9: "The respective differences of STD... are presented in Figure 2." These are different. I guess the latter is correct. The differences were defined between each observation and their minimum, which was observed at a certain azimuth. This definition provides a kink in the graph at the minimum azimuth and then leads to miss-understandings. I suggest that the differences are made between each observation and its average. There is not the land name (POTS) in the body.

P12 Figure 2 It was difficult for me to understand what x- and y-axis labels (Difference of slant delays (mm)) represented (actually, these are not labels for x- and y-axis). Improve locations these labels appear.

P13 Figure 13 This was also made between observations and their minimum. See comment above.

P13 L8: "These values" When were these observed? I guess the observed time were different for each contribution.

P13 L10+1: "the variation range of " The standard deviation and average are better to illustrate such variation statistically. Raw variations may include outliers.

P14 L7: "GPS" Is there any reason to use GPS specifying the US navigation system?

P20 L25: "Note also that ROB_V is consistent with TUO_G." I feel that this sentence is not fear. The authors should list TUO_G at the same sentence (L23).

P22 L16: "orography representation" I guess grid spacings of these three models were different and ALADIN-CZ adopted the smallest. This means the topography of ALADIN-CZ is the most similar to real one. Please show modelled topography of each model and/or modelled altitude in comparison with real one.

P22 L17: "ranging from -3 mm to +7 mm" It is quite difficult to measure these values

from Fig. 8, because there are no scale auxiliary lines for the y axis. Please add the lines not only to Fig. 8 but also other similar figures needed.

P22 L30: "The probable reason ... negative effect of underestimated delays." There is no evidence for this discussion. The authors should show any figures or numbers.

P24 L1: "Chyba! Nenalezen zdroj," Remove these Czech.

P24 section 7.2.2 The authors should reorganize and polish this section, because evidences for discussion in this section are missed by (not presented) or no figures. I would like to point out that one of major error sources in comparison between real and modelled STDs is super refraction in the actual atmosphere. Please examine this point.

P29 This paragraph is not well discussed, because, for instance, there is no figures in the sentence "The biases stay very stable " (L4). It is recommended to show numbers and/or figures in discussion, otherwise, the readers would have to be frustrated to see tables.

P30 This paragraph should be enhanced, because Figure 12 contains much information whereas the discussion is poor.

P31 Conclusions If the authors illustrate discussion sections in connection with these conclusive remarks, it is happy for the readers to see discussion with evidences.

P32 L2: "for STDs to the zenith direction" It is better to use STDs at high elevation angles instead of mapped STDs.

P32 L13: "The impact was" I don't understand what "the impact" illustrates.

P32 L17-18: "The origin was identified as " I did not see any related discussion with this conclusion.

P32 L18-19: "Their values varied at all ... 15 degrees" Is there any discussion on this conclusion?

[Figure]

P33 L15: "hardly as reliable as in needed" Needed for what? State clearly.

Please also note the supplement to this comment:
http://www.atmos-meas-tech-discuss.net/amt-2016-372/amt-2016-372-RC1-
supplement.pdf

———————————————

---

## Referee Comment (RC2) · Anonymous Referee #1 · 27 Jan 2017

In the last decade the assimilation of zenith total delays into numerical weather models became operational at many weather services. At the same time the focus of research shifted to the processing and utilization of slant total delays (STDs). The manuscript presented by Kačmařík et al. describes a comprehensive STD validation study which covers 7 different STD processing strategies and their validation with independent observations. The focus of the manuscript is on the identification of the optimal processing strategy and on the impact of post-fit residuals on the quality of STDs.

This is the most extensive and detailed study in this field which has been presented up to now and it provides a wealth of information. The results are in general well justified and of high significance for GNSS processing and for potential applications as well.

The manuscript is well written and organized. However, some points need clarification and additional information which is important for the reader to better understand different aspects of the validation study. This would require a minor revision.

**General Comments**

Errors
The manuscript compares STD data from different sources: GNSS STDs processed in different ways, raytraced STDs from numerical weather models and STDs obtained from water vapor radiometers. As there is no reliable reference for STD observations, such comparisons provide the difference between two erroneous quantities but not the STD error, i.e. the error with respect to the truth.

At some points of the discussion the authors highlight this aspect but in some cases the standard deviation is regarded as the error of a certain STD product without proper justification. Especially when comparing GNSS STDs with and without residuals the increasing standard deviation due to the residuals is very often regarded as an increasing error. While this might be true in many cases it is not always justified by the analysis.

The manuscript might be improved if this issue is discussed in a paragraph somewhere at the beginning of the analysis and by addressing the corresponding specific comments.

Residuals
The application of residuals is presumably the most important topic in GNSS STD processing. The simple model used in equ. 1 is not sufficient to describe local atmospheric variations in case of severe weather events. Residuals could provide the directional information necessary to locate meteorological phenomena if the GNSS specific errors were below a certain threshold.

In the manuscript the application of residuals is discussed in detail but the analysis

does not lead to a clear recommendation. Regarding the analysis presented in the manuscript the results are well justified. However, the analysis is focused on two month mean values/standard deviations and presumably not the best way to analyze the impact of residuals. Most of the time atmospheric variations are rather smooth and can be described by equ. 1. Under such conditions residuals will probably add some noise to the solution and provide little extra information. In case of severe weather events rather large residuals would be necessary to complement equ. 1 and to locate e.g. convective cells. Under such conditions much larger errors of the residuals could be tolerated. This cannot be analyzed using two month means.

At some points in the manuscript it is mentioned that further studies are required to address this problem but the recommendations how to use residuals remain somewhat indefinite. The presented results could be understood much better if an assessment of the statistical analysis with respect to the application of residuals would be added.

Statistics
The manuscript describes basically a statistical analysis. However, almost nothing is said about the statistical procedures used to analyze the data. At some points bias and standard deviation are used, median, median RMS, median values of biases and standard deviations, mean standard deviations, ... at others. To understand the results it is necessary to describe the statistical analysis and to explain why certain statistical methods are used for a specific analysis.

**Specific Comments**

Abstract, line 18
*Results show generally a very good mutual agreement among all solutions from all the techniques.*
This sentence contradicts the results of the study in some way as the reader gets the impression that all solutions/techniques have almost the same high quality and it

makes no difference which one is used. At the same time it would not be possible to answer the questions raised in the manuscript, i.e. which processing strategy leads to the best STD quality.

The abstract should focus more on the questions which will be answered in the manuscript and on the difficulties to come to a definite conclusion.

Page 3, equ. 1, 2
Equation 1 is essential for the discussion of residuals. Therefore it would be important to discuss the downsides of this approach. Equ. 1 is a rather simple model where all information on elevation is shifted to the mapping functions and variations with the azimuth are described by only two numbers ($G_N$ and $G_E$). Equ. 2 describes a very smooth azimuthal variation which cannot represent the atmospheric state in case of severe weather events. Furthermore, the gradients are temporal means, usually over 1 h. In case of fast moving fronts or convective events the temporal mean can become rather misleading and can lead to an unrealistic azimuth distribution of the STDs.

Using this approach all information provided by GNSS observations is reduced to 3 numbers (ZWD, $G_N$ and $G_E$, assuming that the Saastamoinen ZHD is used) and no directional information survives this process. If it turned out that this is the best way to model atmospheric variations the processing of STDs would be almost meaningless. It would be sufficient to provide these quantities and the user could compute any number of STDs in any direction.

For for the sake of completeness it should be defined how residuals are applied, i.e. equ. 1 + residual.

Page 11, section 4.4
The results presented in this section have presumably be obtained using a numerical weather model. This should be mentioned as the real situation might differ from the model state. Which model was used?

Page 13, line 6,7
*... and corresponding delays in the zenith direction have been computed and mapped using mapping functions presented in Eq. 1 ...*
Why do we need ZTDs to understand fig. 3? It seems that STDs are computed using the weather model and that the differences are mapped to zenith and shown in fig.3.

Page 13, line 12-15
The sentence *Figure 3 confirms ...* sounds somewhat strange and should be rephrased.

Page 14, line 18, 19
What is the *hydrostatic mapping function derived from the NCEP-GFS*?

Page 14, line 19-22
SIWV to STD: Why are hydrostatic horizontal gradients required to convert SIWV to STD? Both observations have been taken in (almost) the same direction and the SHD in this direction should be sufficient. Li, 2015b, describes a way to estimate gradients from different SIWV observations for GNSS gradient validation. Has this also been done?

Page 16 - 21, section 7.1
Section 7.1 is quite large and it would be very beneficial for the reader to divide it into some subsections, e.g. comparison with GFZ, comparison with/without residuals, differences of software parameters, differences depending on elevation.

Page 15, line 18, 19
*Hence the smaller values for these settings, the smaller number of pairs found and the higher standard deviations resulted between GNSS and WVR STDs.*
Shouldn't it be ... **smaller** standard deviations ... ?

Page 16, line 14

*These were observed mainly as systematic errors ranging from -3.6 mm to 0.6 mm.*
Fig. 4 shows differences between the GFZ solution and all other solutions. As long as
the error of the GFZ solution is unknown it's not possible to attribute the differences as
systematic errors.

Page 17, line 8, 9, discussion of pages 17 - 19
*Both comparisons demonstrate systematic errors at a sub-millimetre level over all sta-
tions and solutions.*
Adding residuals to the nonRES solution should lead to a somewhat different bias and
a larger standard deviation, even in case of true, error free residuals. This is due to the
spatial and temporal variability of the atmosphere and not necessarily an error. How-
ever, reading the discussion one gets the impression that smaller biases and standard
deviations are better. This section could be improved by an evaluation of the informa-
tion and potential errors provided by different solutions.

Page 19, line 18, 19
*Surprisingly, the impact of the elevation angle cut-off ($3°$ versus $7°$) resulted in a mini-
mum mean standard deviation below 1 mm, see TUW-3 and TUW-7.*
The impact of low elevation STDs below $7°$ depends considerably on the amount of
data below $7°$. The small impact on bias and SDEV could be due to the small amount
of data or due to the high quality of the data.

Page 21, section 7.2
It would be very beneficial for the reader to start section 7.2 with a short summary of
section 4.1 – 4.3. A short paragraph and a table giving the main parameters of the
weather models and raytracers would be helpful.

Page 26, line 12, 13
*... and it should be noted that the stability on a daily time scale was much better for
GNSS STDs than for NWM ray-traced STDs.*

This is presumably not a problem of the model stability but indicates that the model state deviates from the real atmospheric state for some time/region. This is the usual behavior of weather models which cannot be avoided even if STDs are assimilated.

Page 30, line 2
*Two sets of STDs from the same solution, but ...* Which solution was used in this section?

Page 34, line 7, 8
*... that we are currently not able to remove completely all other effects due to the local troposphere.*
Isn't that misleading? The ideal residual should describe the effect of the local troposphere while all GNSS specific errors should be removed.

**Technical Corrections**

Page 1, line 30
*... part, caused by the atmospheric constituents, and the wet ...*
Shouldn't it be ... the **dry** atmospheric constituents ... ?

Page 10, line 14
$R_d = 287.058$ J/(kg K) = $287.058$ J kg$^{-1}$ K$^{-1}$

Page 11, line 5
Contribution to hydrometeors: 17 mm to ZTD or STD?

Page 11, fig. 1
Fig. 1 needs some improvement: It should be clearly indicated which subplot shows which quantity. The text and the color bars inside the subplots cannot be read.

Page 12, fig. 2
It's rather unusual to provide polar plots with a x and y axis. The angles (azimuth) and the radial axis ($\Delta$STD) should be given.

Page 16, line 8, 9
*Figure 4: Comparison ...* It seems that parts of the figure's caption have accidentally been copied.

Page 16, line 15
*It is particularly ...* What does "It" mean?

Page 22, line 8, p. 25, l. 21, ...
side => site

Fig. 4, 6, 7, 9, 11
These plots show a large number of symbols/lines and could be improved by scaling the y axes according to the min/max values in the plots.

---

## Author Comment (AC1) · 13 Apr 2017

**Interactive comments on "Inter-technique validation of tropospheric slant total delays" by M. Kačmařík et al.**

AUTHOR COMMENTS ON THE CHANGES IN THE MANUSCRIPT DUE TO A RESULTS MISTAKE FOUND IN THE PREVIOUS (REVIEWED) VERSION OF THE MANUSCRIPT

We found a bug in processing which influenced GNSS slant total delays from both TUO solutions (TUO_G and TUO_R) and one ROB (ROB_V) solution based on the VMF1 mapping function. We recomputed all the three affected GNSS STD solutions and consequently all the statistical comparisons presented in the paper. In the new version of the manuscript which also incorporates comments of both reviewers we provide corrected versions of figures (Figures 4, 7, 8, 10, 11, 12) and tables (Tables 4, 5, 7, 8) which were affected by the described mistake in three GNSS STD solutions. The changes are mainly visible in BIAS results in Table 4 and Figure 7, all other outputs were influenced only marginally. We also corrected manuscript text in section 7.1 (sections 7.1.3 and 7.1.4 in the new version of the manuscript) which discussed the previous wrong results of ROB_V and TUO_G/TUO_R solutions.

In the last decade the assimilation of zenith total delays into numerical weather models became operational at many weather services. At the same time the focus of research shifted to the processing and utilization of slant total delays (STDs). The manuscript presented by Kačmařík et al. describes a comprehensive STD validation study which covers 7 different STD processing strategies and their validation with independent observations. The focus of the manuscript is on the identification of the optimal processing strategy and on the impact of post-fit residuals on the quality of STDs.

This is the most extensive and detailed study in this field which has been presented up to now and it provides a wealth of information. The results are in general well justified and of high significance for GNSS processing and for potential applications as well.

The manuscript is well written and organized. However, some points need clarification and additional information which is important for the reader to better understand different aspects of the validation study. This would require a minor revision.

**General comments**

Errors

The manuscript compares STD data from different sources: GNSS STDs processed in different ways, raytraced STDs from numerical weather models and STDs obtained from water vapor radiometers. As there is no reliable reference for STD observations, such comparisons provide the difference between two erroneous quantities but not the STD error, i.e. the error with respect to the truth.

At some points of the discussion the authors highlight this aspect but in some cases the standard deviation is regarded as the error of a certain STD product without proper justification. Especially when comparing GNSS STDs with and without residuals the increasing standard deviation due to the residuals is very often regarded as an increasing error. While this might be true in many cases it is not always justified by the analysis.

The manuscript might be improved if this issue is discussed in a paragraph somewhere at the beginning of the analysis and by addressing the corresponding specific comments.

We agree with your point of view. We haven't provided an extra paragraph in the beginning of the analysis but describe and discuss this topic in the beginning of section 8 and we also addressed all your specific comments in the manuscript.

Although the truth is not accessible in our as well as in similar studies, besides technique inter-comparisons, we have particularly focused on assessing variants without, with raw and cleaned residuals at the dual stations (section 8). In such case the assumption of similar residuals due to the tropospheric effects is expected at nearby located stations and slant delays should be thus zero, in theory.

Residuals

The application of residuals is presumably the most important topic in GNSS STD processing. The simple model used in equ. 1 is not sufficient to describe local atmospheric variations in case of severe weather events. Residuals could provide the directional information necessary to locate meteorological phenomena if the GNSS specific errors were below a certain threshold.

The residual term (RES) was added to Eq.1 and the text in Chapter 3 was adapted and re-ordered accordingly.

In the manuscript the application of residuals is discussed in detail but the analysis does not lead to a clear recommendation. Regarding the analysis presented in the manuscript the results are well justified.

However, the analysis is focused on two month mean values/standard deviations and presumably not the best way to analyze the impact of residuals. Most of the time atmospheric variations are rather smooth and can be described by equ. 1. Under such conditions residuals will probably add some noise to the solution and provide little extra information. In case of severe weather events rather large residuals would be necessary to complement equ. 1 and to locate e.g. convective cells. Under such conditions much larger errors of the residuals could be tolerated. This cannot be analyzed using two month means.

At some points in the manuscript it is mentioned that further studies are required to address this problem but the recommendations how to use residuals remain somewhat indefinite. The presented results could be understood much better if an assessment of the statistical analysis with respect to the application of residuals would be added.

Please, see a new version of section 8. We have identified days with high daily variations of cleaned post-fit residuals (corresponding to severe weather occurrences) and studied separately results for these days and days with low variation of post-fit residuals. Here we present Fig. 1 showing daily RMS of cleaned post-fit residuals at elevation angles of 10 and 30 ° at individual GNSS stations forming dual stations.

[Figure]

Fig. 1: RMS of clean post-fit residuals at elevation angles of 10 and 30 ° for individual days of benchmark at all GNSS stations forming dual stations

Statistics

The manuscript describes basically a statistical analysis. However, almost nothing is said about the statistical procedures used to analyze the data. At some points bias and standard deviation are used, median, median RMS, median values of biases and standard deviations, mean standard deviations, ... at others. To understand the results it is necessary to describe the statistical analysis and to explain why certain statistical methods are used for a specific analysis.

We changed Figure 9 and, currently, there are only following statistical parameters used: bias, standard deviation, median values of biases and standard deviations (made over all stations or solutions). The only exception is abstract where we state mean values. We have shortly broadened the explanation of using median values of biases and standard deviations in the Section 6.

**Specific Comments**

Abstract, line 18

*Results show generally a very good mutual agreement among all solutions from all the techniques.*

This sentence contradicts the results of the study in some way as the reader gets the impression that all solutions/techniques have almost the same high quality and it makes no difference which one is used. At the same time it would not be possible to answer the questions raised in the manuscript, i.e. which processing strategy leads to the best STD quality.

The sentence was modified – we wanted to sum up the results into a single sentence and actually we found a reasonable agreement among most of the solutions from all the techniques.

The abstract should focus more on the questions which will be answered in the manuscript and on the difficulties to come to a definite conclusion.

The last part of the abstract was significantly modified.

Page 3, equ. 1, 2

Equation 1 is essential for the discussion of residuals. Therefore it would be important to discuss the downsides of this approach. Equ. 1 is a rather simple model where all information on elevation is shifted to the mapping functions and variations with the azimuth are described by only two numbers (GN and GE). Equ. 2 describes a very smooth azimuthal variation which cannot represent the atmospheric state in case of severe weather events. Furthermore, the gradients are temporal means, usually over 1 h. In case of fast moving fronts or convective events the temporal mean can become rather misleading and can lead to an unrealistic azimuth distribution of the STDs.

Using this approach all information provided by GNSS observations is reduced to 3 numbers (ZWD, GN and GE, assuming that the Saastamoinen ZHD is used) and no directional information survives this process. If it turned out that this is the best way to model atmospheric variations the processing of STDs would be almost meaningless. It would be sufficient to provide these quantities and the user could compute any number of STDs in any direction.

For for the sake of completeness it should be defined how residuals are applied, i.e. equ. 1 + residual.

We have edited whole Section 3 including equation 1.

Page 11, section 4.4

The results presented in this section have presumably be obtained using a numerical weather model. This should be mentioned as the real situation might differ from the model state. Which model was used?

Text added in the manuscript (ALADIN-CZ NWM has been used to estimate the Hydrostatic, Wet and Hydrometeor contributions to slant delays.)

Page 13, line 6,7

*... and corresponding delays in the zenith direction have been computed and mapped using mapping functions presented in Eq. 1 ...*

Why do we need ZTDs to understand fig. 3? It seems that STDs are computed using the weather model and that the differences are mapped to zenith and shown in fig.3.

No new ZTDs have been computed. Only STDs have been mapped in the zenith to avoid the effect of the elevation and to look at the same order of magnitude of delays. Text modified in the manuscript.

Page 13, line 12-15

The sentence *Figure 3 confirms* ... sounds somewhat strange and should be rephrased.

Text modified in the manuscript

Page 14, line 18, 19

What is the *hydrostatic mapping function* derived from the NCEP-GFS?

We make use of GFS NCEP data to derive the hydrostatic MF. For details see below reference which we added into text.

Douša, J., Dick, G., Kačmařík, M., Brožková, R., Zus, F., Brenot, H., Stoycheva, A., Möller, G., and Kaplon, J.: Benchmark campaign and case study episode in central Europe for development and assessment of advanced GNSS tropospheric models and products, Atmos. Meas. Tech., 9, 2989-3008, doi:10.5194/amt-9-2989-2016, 2016.

Page 14, line 19-22

SIWV to STD: Why are hydrostatic horizontal gradients required to convert SIWV to STD? Both observations have been taken in (almost) the same direction and the SHD in this direction should be sufficient. Li, 2015b, describes a way to estimate gradients from different SIWV observations for GNSS gradient validation. Has this also been done?

We approximate WVR STD by

$STD = SIWV * PI + m\_h * Z\_h + m\_g * (cos(a) N\_h + sin(a) E\_h)$
$m\_h$ … hydrostatic MF (from GFS NCEP, see the comment above)
$Z\_h$ … zenith hydrostatic delay (using Saastamoinen and in-situ meteorological observations)
$m\_g$ … gradient MF
$N\_h, E\_h$ … hydrostatic north and east gradient (from GFS NCEP, see the comment above)
Note that in general $N\_h$ & $E\_h$ are not zero. Conversely, we would approximate GPS SIWV by
$SIWV = [ STD - m\_h * Z\_h - m\_g * (cos(a) N\_h + sin(a) E\_h)] / PI$

To answer the second question: no, we did not compute gradients from WVR measurements.

Page 16 - 21, section 7.1

Section 7.1 is quite large and it would be very beneficial for the reader to divide it into some subsections, e.g. comparison with GFZ, comparison with/without residuals, differences of software parameters, differences depending on elevation.

The text in Section 7.1 was divided into four subsections 7.1.1 – 7.1.4.

Page 15, line 18, 19

Hence the smaller values for these settings, the smaller number of pairs found and the higher standard deviations resulted between GNSS and WVR STDs. Shouldn't it be ... **smaller** standard deviations ... ?

Yes, smaller is right, corrected in the manuscript.

Page 16, line 14

*These were observed mainly as systematic errors ranging from -3.6 mm to 0.6 mm.*

Fig. 4 shows differences between the GFZ solution and all other solutions. As long as the error of the GFZ solution is unknown it's not possible to attribute the differences as systematic errors.

Manuscript was modified, now the term bias is used.

Page 17, line 8, 9, discussion of pages 17 - 19

*Both comparisons demonstrate systematic errors at a sub-millimetre level over all stations and solutions.*

Adding residuals to the nonRES solution should lead to a somewhat different bias and a larger standard deviation, even in case of true, error free residuals. This is due to the spatial and temporal variability of the atmosphere and not necessarily an error. However, reading the discussion one gets the impression that smaller biases and standard deviations are better. This section could be improved by an evaluation of the information and potential errors provided by different solutions.

We agree and therefore modified the text of section 7.1.2 however didn't updated it to add more information in regard of the last sentence of the comment.

Page 19, line 18, 19

*Surprisingly, the impact of the elevation angle cut-off (3 versus 7) resulted in a minimum mean standard deviation below 1 mm, see TUW-3 and TUW-7.*

The impact of low elevation STDs below 7 depends considerably on the amount of data below 7. The small impact on bias and SDEV could be due to the small amount of data or due to the high quality of the data.

Yes, we agree with your statement. We want to add that although a cut-off elevation angle 3 ° was used for the TUW-3 solution, STDs from below 7 ° didn't enter the validation. Therefore, the difference between TUW-3 and TUW-7 solutions comes mainly from estimated horizontal gradients. We edited the manuscript in these regards.

Page 21, section 7.2

It would be very beneficial for the reader to start section 7.2 with a short summary of section 4.1 – 4.3. A short paragraph and a table giving the main parameters of the weather models and raytracers would be helpful.

We slightly modified text in the manuscript (beginning of section 4 and section 7.2). We don't want to repeat the information given in the beginning of section 4, therefore we refer the reader to go back to section 4 in the beginning of section 7 to refresh the information about NWM STD solutions.

Page 26, line 12, 13

*... and it should be noted that the stability on a daily time scale was much better for GNSS STDs than for NWM ray-traced STDs.*

This is presumably not a problem of the model stability but indicates that the model state deviates from the real atmospheric state for some time/region. This is the usual behavior of weather models which cannot be avoided even if STDs are assimilated.

We agree, the sentence was rewritten.

Page 30, line 2

*Two sets of STDs from the same solution, but ...* Which solution was used in this section?

Corrected, an explanation was given.

Page 34, line 7, 8

*... that we are currently not able to remove completely all other effects due to the local troposphere.*

Isn't that misleading? The ideal residual should describe the effect of the local troposphere while all GNSS specific errors should be removed.

Corrected, the sentence was rephrased.

**Technical Corrections**

Page 1, line 30

*... part, caused by the atmospheric constituents, and the wet ...* Shouldn't it be ... the **dry** atmospheric constituents ... ?

Corrected.

Page 10, line 14

$R_d = 287.058$ J/(kg K) $= 287.058$ J kg$^{-1}$ K$^{-1}$

Corrected.

Page 11, line 5

Contribution to hydrometeors: 17 mm to ZTD or STD?

In the zenith, corrected in the manuscript.

Page 11, fig. 1

Fig. 1 needs some improvement: It should be clearly indicated which subplot shows which quantity. The text and the color bars inside the subplots cannot be read.

Figure was modified.

Page 12, fig. 2

It's rather unusual to provide polar plots with a x and y axis. The angles (azimuth) and the radial axis ($\Delta$STD) should be given.

Figure was modified.

Page 16, line 8, 9

*Figure 4: Comparison* ... It seems that parts of the figure's caption have accidentally been copied.

Corrected

Page 16, line 15

*It is particularly* ... What does "It" mean?

The sentence relates to the previous one describing similarity of GFZ and GOP results.

Page 22, line 8, p. 25, l. 21, ...

side => site

No, the word side is correct. We wanted to say that the bias between POTM and POTS stations comes from GNSS data processing, not from NWM derived STDs.

Fig. 4, 6, 7, 9, 11

These plots show a large number of symbols/lines and could be improved by scaling the y axes according to the min/max values in the plots.

We adapted the y scales in figures 4, 7, 9, 11. We kept the scales in figures 5+6, 8, 10, 14 since we want to keep an identical y scale for all similar figures to make them consistently comparable with each other.

This manuscript describes inter-comparisons of slant total delays (STDs) derived from GNSS solutions, numerical weather models (NWM), and radiometers (WVR). In comparisons between GNSS software, the authors found most of them show good agreements with each other. Moreover, they recommend no use of raw post-fit residuals whereas STDs without residuals and with cleaned residuals are better. As for NWM and WVR, the authors concluded they are not reliable due to their large errors. I roughly agree with their conclusions. However, since I have to point out some weaknesses in this manuscript, it should be published after major revisions.

Major comments 1) Residuals STD is decomposed with ZTD, gradients (G) and residuals. Since Equation 1 only represents the first two terms, the residual term should be added. Since the horizontal scales of ZTD, G, and residuals are 500, 50, and 5 km (Shoji et al. 2004), it is important to add residuals into STDs when convective activities are considered in any studies using STDs like this manuscript. From the view point of this, the authors have no chance to avoid residuals in processing STDs. In addition, residuals should be cleaned as pointed by Shoji et al. (2004). Therefore, I don't agree that "the usage of the information content from the post-fit residuals for the reconstruction of the STDs remains an open question" in this study (L5 P5). The authors should re-consider the effect of post-fit residuals in their formulation and re-organize this manuscript from the view point that they really need to investigate the effect of post-fit residuals in this study.

The residual term (RES) was added to Eq.1 and the text in Chapter 3 was adapted and re-ordered accordingly. Regarding the post-fit residuals investigation: please see following sections of the revised manuscript: 8, 9.

2) Comparison in the zenith direction. The authors compared STDs in the zenith direction using mapping functions. Since these functions were made statistically (excluding gradient on the day), I recommend the authors to use STDs only in high elevation angles (> 60 or 70 degree) for the comparisons. This is especially useful in comparison of GNSS vs WVR, because it is able to avoid errors of surface pressure gradient in calculating STDs from WVR.

For comparisons in the zenith direction we didn't use any real mapping function, we used only a simple 1/sin(ele) to normalize all the STDs. We didn't want to reconstruct original zenith parameters for each STD, but we wanted to compare all the variable STDs as a one unit. Mapping slant observations to the zenith direction is a standard approach in validation studies (i.e. Bender et al. (2008), Deng et al. (2011), Shang-Guan et al. (2015)) and the applying of the same mapping functions/factors for compared slants corresponds to the residuals 'normalization' with impact representing a second-order effect. We don't think that using STDs coming only from high elevation angles, as you propose, would allow us to validate the overall quality of STDs.

Regarding the GNSS vs WVR comparisons: We know that WVR measurements at low elevation angles can be of low quality, however, we didn't want to exclude them completely from the validation. Therefore, we selected a compromise of 15° elevation cut-off for these comparisons. Most of the so far presented studies used even lower elevation cut-off for WVR data (i.e. Bender et al. (2008) used 5 ° cut-off, Shang-Guan et al. (2015) used 7 ° cut-off, Braun et al. (2001) and Braun et al. (2002) used 10 ° cut-off, Deng et al. (2011) used 15 ° cut-off, Li et al. (2015a) used 20 ° cut-off since WVR wasn't measuring below this angle). We added dry horizontal gradients derived from NWM to WVR measurements to avoid errors of surface pressure gradients, please, see Section 5 describing the reconstruction of STDs from original SIWV WVR measurements.

3) Comparison with NWM There are three models appeared in this manuscript: ERAInterim, NCEP GFS, and ALADIN-CZ. To help the readers understand the discussion on this topic, please describe more settings on these models.

- Within these, only ALADIN is a regional model, and others are global. State their general characteristics more.

  We added more information about the models to the beginning of chapter 4 together with references to other papers/relevant web pages.

- Do all of these models assimilate GNSS data for their initial conditions or not?

  None of these models assimilate data from ground based stations. We added this information to manuscript (beginning of chapter 4)

- How large are their grid spacings?

  The horizontal resolution for both ERA and NCEP GFS is 1°, for ALADIN-CZ it is 4.7 km (see the beginning of section 4).

- ERA-Interim is the ECMWF re-analysis data produced 6 hourly. I guess GFS is 6 hourly operational analysis data at NCEP. What is ALADIN-CZ?

  With regards to ERA and NCEP: the assumption is correct. ALADIN-CZ is also 6 hourly operational analysis with forecasts for 0, 1, 2, 3, 4, 5, 6 hours. Information added to manuscript.

- Does ALADIN-CZ have large domain enough to produce STDs? I concern that STDs at low elevation angles might penetrate the lateral boundary of the model and need special treatment like STDs over the top of the model.

  There is no problem in ALADIN-CZ domain which is roughly 4,000*3,000 km, see figure 3 in Douša et al. (2016).

- ALADIN-CZ may be a cloud-permitting model (this depends on its resolution) with explicit cloud microphysics. In this case, it is possible to calculate STDs with hydrometer effect. Is this right? If yes, I suggest to do this (see the major comment 5).
  This was done and is described in section 4.4.

P7 section 4 There are two error sources in this comparison; STD solutions and NWMs. I suggest the authors to employ single STD solution with three NWMs and then compare the results with observed STDs. This makes error sources reduced single (only NWM) and discussion much easier (section 7.2).

We agree with the suggestion, therefore the Figure 9 was adopted and now is showing results of NWM versus GNSS GFZ solution. We rewrote the text in the chapter with the description of this figure.

4) Figures and discussion Although there are many graphs appeared in this manuscript, some of them are not appropriate for discussion. For instance, though Figure 12 displays 30 lines in 12 panels, the authors made a discussion only in single paragraph (P30 L10). Another example. Although the authors showed small number of figures in connection with GNSS versus NWM comparison, they discussed many points (stations) without figures in section 7.2. I recommend to re-organize discussion and figures.

Our goal was to balance the already significant length of the paper while giving as much as possible information - therefore for some of the results we haven't provided figures/tables. However, we try to rectify it in this document and we provide additional figures (Fig. 2, Fig. 3) to prove our statements from the text. They can be found in the text below where we address specific comments.

5) Assessment of components in the atmosphere Although the discussion on the effect of each component of the atmosphere (section 4.4) is important, the authors did not show any conclusion. I suggest to examine the same effect using NWMs additionally and illustrate useful information.

We struggle to understand this major comment. In section 4.4 we present results from NWM ALADIN-CZ.

**Minor comments**

P1 L21: "between GNSS a NWM" Reword to "between GNSS and NWM".

Corrected

P1 L29: "along his path" Reword to "along the path"

Corrected

P4 L8: "was operating only" Reword to "was operated only" (?)

Corrected

P6 L91: "three variants of the solution" I don't think that it is worth to examine "nonRES" case in this paper. See my major comment.

We don't think we should completely skip "nonRES" solutions from the validation. It allowed us to evaluate and quantify the impact of post-fit residuals.

P8 L4: "mix ratio of liquid" mixing ratio of liquid

Corrected

P10 L24: "The contribution of water – neglected in the total delay." As I mentioned in my major comments, I suggest the author to examine these contributions.

We examined these contributions in section 4.4. ALA/BIRA solution considered (added) these impacts in its STDs, ALA/WUELS solution did not.

P11 Figure 1 Enlarge the land names.

Figure modified with increase of the size of the name of the stations (see attachment)

P12 L4: "Figure 2 shows simulated STDs" P12 L9: "The respective differences of STD. . . are presented in Figure 2." These are different. I guess the latter is correct. The differences were defined between each observation and their minimum, which was observed at a certain azimuth. This definition provides a kink in the graph at the minimum azimuth and then leads to miss-understandings. I suggest that the differences are made between each observation and its average.

Yes, this is right.

There is not the land name (POTS) in the body

Now POTS is visible in Fig. 1

P12 Figure 2 It was difficult for me to understand what x- and y-axis labels (Difference of slant delays (mm)) represented (actually, these are not labels for x- and y-axis). Improve locations these labels appear.

Figure improved.

P13 Figure 3 This was also made between observations and their minimum. See comment above.

Yes, right.

P13 L8: "These values" When were these observed? I guess the observed time were different for each contribution.

Minimal values obtained during the whole period of the benchmark campaign (text added into manuscript).

P13 L10+1: "the variation range of " The standard deviation and average are better to illustrate such variation statistically. Raw variations may include outliers.

Because we look at severe weather situation, this is interesting for use to know the variation range, this is an indicator of the severity of the meteorological events. We are not sure if we understand your comment well.

P14 L7: "GPS" Is there any reason to use GPS specifying the US navigation system?

Yes, since the WVR is able to track only GPS satellites, not GLONASS or any other GNSS.

P20 L25: "Note also that ROB_V is consistent with TUO_G." I feel that this sentence is not fear. The authors should list TUO_G at the same sentence (L23).

Read, please, our comments in the beginning of the document regarding a mistake in ROB_V and TUO_G (TUO_R) solutions. Whole paragraph of manuscript including this sentence was edited (deleted).

P22 L16: "orography representation" I guess grid spacings of these three models were different and ALADIN-CZ adopted the smallest. This means the topography of ALADINCZ is the most similar to real one. Please show modelled topography of each model and/or modelled altitude in comparison with real one.

We present a table with differences in heights between GNSS and NWM models ERA-Interim and ALADIN-CZ for all individual GNSS reference stations. We interpolated the NWM heights from four neighboring grid points. For NCEP GFS model it is not possible to provide differences in height since only pressure level fields were used to estimate the STDs and no surface layer field.

| GNSS station | GNSS – ERA-Interim [m] | GNSS – ALADIN-CZ [m] |
|---|---|---|
| GOPE | 169.6 | 177.7 |
| KIBG | 674.0 | 328.4 |
| LDB2 | 81.0 | 60.5 |
| POTS | 65.6 | 75.5 |
| SAAL | 754.3 | 274.6 |
| WTZR | 187.7 | 127.5 |

From these altitude differences, we expect to find the largest discrepancies between the GNSS and NWM derived STDs at the station KIBG and SAAL and indeed this is what we see in Fig. 8. The reason is that the representative error and the interpolation/extrapolation error (in the ray-tracing algorithm the refractivity from the surrounding grid points is interpolated/extrapolated to the desired location) are largest at these stations. We are not confident with the ALADIN-CZ STD simulations for KIBG and SAAL. The reason is because the hydrostatic and the wet component are not well simulated. This can be due to a problem of ALADIN-CZ for simulating small-scale structures at the good locations (ERA-interim simulates structures with a larger scale). Then this can explain strong deviation/anisotropy between STD observed and simulations.

P22 L17: "ranging from -3 mm to +7 mm" It is quite difficult to measure these values from Fig. 8, because there are no scale auxiliary lines for the y axis. Please add the lines not only to Fig. 8 but also other similar figures needed.

Done, Figs 4 – 11, 14 were modified.

P22 L30: "The probable reason . . . negative effect of underestimated delays." There is no evidence for this discussion. The authors should show any figures or numbers.

We edited the text in the manuscript to add a link to chapter 7.2.2 and we also provide here Fig. showing comparison between GNSS GFZ solution and all four NWM based STD solutions including ALA/WUELS for station KIBG where GNSS GFZ – ALA/WUELS reach 330 mm of absolute bias and 270 mm of absolute standard deviation.

P24 L1: "Chyba! Nenalezen zdroj," Remove these Czech.

Corrected.

P24 section 7.2.2 The authors should reorganize and polish this section, because evidences for discussion in this section are missed by (not presented) or no figures. I would like to point out that one of major error sources in comparison between real and modelled STDs is super refraction in the actual atmosphere. Please examine this point.

Below, we provide two figures (Fig. , Fig. ) which are variants of Figure 9 from the manuscript showing comparisons between GNSS GFZ solution and NWM STD solutions in the slant direction. Please note that both these figures use different y axis scales than Figure 9 included in the manuscript and show results for different stations.. Fig. proves our discussion regarding poor quality of ALA/WUELS STD solution visible mainly at low elevation angles. Fig. show results for station WTZZ and support our statements regarding sudden increases and decreases of standard deviation values at various elevation angles at stations WTZR, WTZS and WTZZ occurring mainly in case of ALA/BIRA solution. Plots for WTZR and WTZS stations show completely the same behavior (not presented here).

To answer the super refraction issue: We consider ground-based stations and restrict ourselves to elevation angles above 3°. We do not find any hint in literature that super refraction (the N-gradient exceeds a critical value) is a problem with such set up (ground-based station and elevation angles well above 3°).

[Figure]

Fig. 2: Comparison of NWM-based solutions (ALA/BIRA, ERA/GFZ, GFS/GFZ, ALA/WUELS) against GNSS GFZ solution at station KIBG, in the slant direction.

[Figure]

Fig. 3: Comparison of NWM-based solutions (ALA/BIRA, ERA/GFZ and GFS/GFZ) against GNSS GFZ solution at station WTZZ, in the slant direction.

P29 This paragraph is not well discussed, because, for instance, there is no figures in the sentence "The biases stay very stable " (L4). It is recommended to show numbers and/or figures in discussion, otherwise, the readers would have to be frustrated to see tables.

The sentence is related to Tables 7 and 8 which are introduced just in the previous sentence. Therefore we don't think it is necessary to reference them again.

P30 This paragraph should be enhanced, because Figure 12 contains much information whereas the discussion is poor.

Please, read the revised Section 8 with the dual stations results – the description of mentioned figure is now in part 8.1.

P31 Conclusions If the authors illustrate discussion sections in connection with these conclusive remarks, it is happy for the readers to see discussion with evidences.

We have rewritten the conclusion chapter as well as provided extra results in this document (see figures above).

P32 L2: "for STDs to the zenith direction" It is better to use STDs at high elevation angles instead of mapped STDs.

Please, see our reply to your Major comment number 2.

P32 L13: "The impact was" I don't understand what "the impact" illustrates.

It meant the impact of adding post-fit residuals to STDs, the manuscript was corrected.

P32 L17-18: "The origin was identified as " I did not see any related discussion with this conclusion.

See Fig.  in this document and our reactions to your comment for P22 L30 and P24 section 7.2.2.

P32 L18-19: "Their values varied at all . . . 15 degrees" Is there any discussion on this conclusion?

There is a short discussion in the section 7.2.2, we rephrased the sentence in the conclusion.

P33 L15: "hardly as reliable as in needed" Needed for what? State clearly.

The sentence was rephrased.

Please also note the supplement to this comment:

http://www.atmos-meas-tech-discuss.net/amt-2016-372/amt-2016-372-RC1-supplement.pdf

We found the supplement to be completely the same as the provided review and, hopefully, we answered all the comments/questions.